# The transient sensitivity of sea level rise

Aslak Grinsted[1] and Jens Hesselbjerg Christensen[1,2]

[1]Physics of Ice, Climate and Earth, Niels Bohr Institute, University of Copenhagen, Denmark.
[2]NORCE Climate, Bergen, Norway

*Correspondence to*: Aslak Grinsted (aslak@nbi.ku.dk)

**Abstract.** Recent assessments from the Intergovernmental Panel on Climate Change (IPCC) imply that global mean sea level is unlikely to rise more than about 1.1m within this century, but will increase further beyond 2100. Even within the most intensive future anthropogenic greenhouse gas emission scenarios are higher levels assessed to be unlikely. However, some studies conclude that considerably greater sea level rise could be realized, and a number of experts assign a substantially higher

likelihood of such a future. To understand this discrepancy, it would be useful to have scenario independent metrics that can be compared between different approaches. The concept of a transient climate sensitivity has proven to be useful to compare the global mean temperature response of climate models to specific radiative forcing scenarios. Here, we introduce a similar metric for sea level response. By analyzing mean rate of change in sea level (not sea level itself), we identify a near linear relationship with global mean surface temperature (and therefore accumulated carbon dioxide emissions) in both model

projections, and in observations on a century time scale. This motivates us to define the 'Transient Sea Level Sensitivity' as the increase in the sea level rate associated with a given warming in units of m/century/K. We find that future projections estimated on climate model responses fall below extrapolation based on recent observational records. This comparison suggests that the likely upper level of sea level projections in recent IPCC reports would be too low.

## 1 Introduction

Our planet is warming as anthropogenic emissions are increasing the atmospheric concentration of carbon dioxide. This warming causes sea levels to rise as oceans expand and ice on land melts. A perturbation in greenhouse gas concentrations changes the balance of energy fluxes between the atmosphere and the ocean surface, and the balance of mass fluxes to and from glaciers and ice sheets. However, the oceans and ice sheets are vast and it takes centuries to heat the oceans, and millenia for ice sheets to respond and retreat to a new equilibrium (Clark et al. 2018; Li et al., 2013; De Conto and Pollard, 2016;

Oppenheimer et al. 2019). In this sense the ice sheets and oceans have a large inertia: An increase in forcing result in a long-term commitment to sea level rise. Simulations by Clark et al. (2018) indicate an equilibrium sea level sensitivity of ~2m/100 GtC emitted $CO_2$. The equilibrium sensitivity can be compared to paleo-data (e.g. Foster and Rohling, 2013). Initially the response to a perturbation in forcing is a flux imbalance, i.e. a change in the rate of sea level rise. Hence, sea level rise by 2100 does not immediately reflect the temperature in 2100, instead the entire pathway since the forcing change was introduced is

important. We therefore expect 21st century sea level rise to better correlate with the century averaged temperature than

temperature itself by 2100. Following this, we therefore propose to linearize the relationship between average rate of sea level rise and temperature increase representing the entire preceding century. The slope of this relationship then expresses how sensitive sea level is to century time-scale warming, and we will refer  to it as the *transient sea level sensitivity* (TSLS). The intercept - where the sea level rate of change is zero - we interpret as a *balance temperature*. The relationship between the temperature and the rate of sea level rise has previously been noted (e.g. Warrick and Oerlemans, 1990), and has been used to motivate semi-empirical models of sea level rise (Rahmstorf, 2007; Grinsted et al. 2010; Church et al. 2013; Kopp et al., 2016; Mengel et al., 2016). A key assumption behind such semi-empirical model projections is that the sensitivity implied by historical records is stationary and hence can be extrapolated into the future. However, there may be processes that can cause future sensitivity to be different from the past (Church et al., 2013). These changes can broadly be categorized as being due to a non-linear response to forcing, or due to a non-stationary response where the response depends on the state of the system. E.g. the sensitivity of small glaciers to warming will depend on how much glacier mass there is left to be lost, and we therefore expect this to have a non-stationary response. Nature is complex and will be both non-linear and non-stationary, and this places limits on extrapolation. Regardless, the sea level response can always be characterized using the TSLS metric, and we can compare and contrast different estimates.

## 2 Reflections on the method

Sea level projections in the IPCC Fifth Assessment Report (AR5; Church et al., 2013), and the Special Report on the Ocean and Cryosphere in a Changing Climate (SROCC; Oppenheimer et al., 2019) are unfortunately not accompanied by hindcasts using the same model framework used for projections. It is therefore impossible to verify that these models can reproduce historical sea level rise. We can, however, compare the TSLS of model projections to the TSLS implied by historical records, and this can serve as a reality check. We have to keep in mind that TSLS potentially can change over time, and that a comparison between different periods cannot be as conclusive. We therefore appeal that future sea level based on modelling are not only used for projections but also include results based on model hindcasts. Ice sheets and ocean heat content has multi century response times and this can lead to model drift if the model is not perfectly initialized. To inform about the future, it is therefore a necessity but not sufficient that model can reproduce the total sea level rise over the 20[th] century. It is critical that sea level models also have sensitivities that are compatible with observations. We therefore propose that the historical TSLS should be used as an emergent constraint of sea level models.

Frederikse et al. (2020) find multi-decadal variability in the relative contributions of the major sea level contributors over the 20[th] century. In recent years the contribution from ice melt has increased relative to that from thermal expansion. We also expect the individual major sea level contributors to have different sensitivities to warming. One might be misled to conclude that TSLS must be changing substantially already. Here, we demonstrate that even in a completely linear world we would

expect to have the budget to be changing over time (see Figure 1). For illustrative purposes we construct a simple linear model where global sea level rise ($\dot{S}$) only has two contributors: ice mass loss ($\dot{M}$) and thermal expansion ($\dot{E}$). We write:

$$\dot{S} = \dot{M} + \dot{E}.$$

These two contributions each respond linearly to warming.

$$\dot{M} = a_M T + b_M$$
$$\dot{E} = a_E T + b_E$$

We insert and get a linear model for the sea level rate:

$$\dot{S} = (a_M + a_E)T + b_M + b_E$$

The proportion of sea level rise due to ice melt becomes

$$\frac{\dot{M}}{\dot{S}} = \frac{a_M T + b_M}{(a_M + a_E)T + b_M + b_E} \ .$$

This is not generally constant in T (see Figure 1), demonstrating that a changing proportion of ice melt does not necessarily imply a changing sensitivity to warming. Church et al. (2013) note that it is very likely that ice-sheet dynamical changes have contributed only a small part of the historical sea level rise, implying that semi-empiric models are unlikely to be able predict

a large future contribution. The fact that ice dynamical changes have only been a minor contributor historically, while we expect it to play an increasingly important role in the future does not imply that TSLS cannot be close to stationary.

## 3 Data

Here we restrict our analysis to published estimates of the Global Mean Sea Level (GMSL) rate. We use three estimates of the historical rate: 1) the tide gauge record (TG) for the period 1900-1990 (Dangendorf et al., 2017); 2) the satellite-altimetry

record (Sat; Ablain et al., 2019) from 1993-2017; 3) a reconstruction for the 1850-1900 pre-industrial period (PI; Kopp et al., 2016). The corresponding temporally averaged temperature anomalies and uncertainties are calculated from the HADCRUT4 observationally based ensemble of Global Mean Surface Temperature (GMST) reconstructions (Morice et al., 2012). We follow IPCC AR5 and use a 1986-2005 baseline for temperature anomalies to avoid introducing additional uncertainties from in re-baselining the IPCC assessed projections. The historical estimates are compared to the projected sea level rate and

temperature from 2000-2100 from two recent IPCC reports for a range of scenarios: the AR5 (Church et al., 2013), and the SROCC (Oppenheimer et al., 2019). Finally, we show the results of an expert elicitation (Bamber et al., 2019) which pertain to scenarios with 2°C and a 5°C warming by 2100 relative to the pre-industrial. These estimates are shown in Figure 2.

## 4 Methods

The relationship between temperature and GMSL rate is estimated for each group of points using linear regression. The three

observational estimates of both temperature and sea level rate (Figure 2, black) are uncertain. We take the uncertainties to be

independent as the three estimates are sourced from separate studies using different data sources, different methods, and are well separated in time. We assume independent gaussian errors which we propagate to our estimates of the line parameters listed in Table 1 using Monte Carlo sampling. Uncertainties in the projections assessed in AR5 and SROCC are specified as a central estimate and a likely range for both temperature and sea level (Church et al., 2013; Oppenheimer et al. 2019; Mastrandea

et al., 2010). The IPCC sources do not provide information on the uncertainty covariance between projections of temperature and sea level. However, we observe that the upper and lower likely limits of temperature paired with the corresponding limit of sea level falls very close to the curve between central estimates (see Figure 2). This indicates that there may well be a very high degree of covariance. For simplicity, we therefore assume full covariance between uncertainties in projected temperature and projected sea level, and depict this using the slanted error bars displayed in Figure 2. This assumption allows us to derive

the upper and lower limit of the likely TSLS range by fitting a line to the corresponding limit of sea level projections. Similarly, we assume covariance between the elicitation derived uncertainties of the two warming scenarios.

Table 1 reports several estimates of TSLS, and we want to understand if each is substantially different to the corresponding observational estimate considering the uncertainties. We therefore test if the absolute difference is larger than zero considering

uncertainties in both estimates, using a standard two-tailed hypothesis test assuming normality.

We show the total cumulated anthropogenic $CO_2$ emissions associated with a given temperature as a secondary horizontal axis in Figure 2 (IPCC, 2013; Meinshausen et al., 2011). We established this relationship using both historical data, and the mid-range temperature projections for the RCP scenarios, and thus do not account for uncertainties in the e.g. climate sensitivity.

The cumulated emission and temperatures were averaged over the same time intervals.

## 5 Results

The estimates of the temporal average rate of sea level rise against corresponding temporal average of GMST from a variety of sources are shown in Figure 2. The AR5 and SROCC projected rate of sea level rise over the 21st century from different scenarios show a close correspondence with projected temperatures (Figure 2, red and blue). The Pearson correlations are

above 0.98 with p<0.001 in a two-tailed test for both AR5 (N=15) and SROCC (N=9), where N is three times the number of scenarios as we include the lower, mid, and upper likely estimates from the reports. We fit straight lines to these projections, and the slope gives a TSLS of $0.27^{+0.03}_{-0.01}$ m/century/K for AR5, and $0.39^{+0.04}_{-0.03}$ m/century/K for the models assessed in SROCC (Table 1). The historical rate of sea level rise in three different periods (PI, TG, and Sat) also show a close relationship to warming (Figure 2, black) with a correlation coefficient of 0.998 (N=3; p<0.05). From this we estimate a TSLS of

$0.40\pm0.05$ m/century/K. Finally, we represent the results of expert elicitation of 21st century sea level rise under two different warming scenarios (Bamber et al. 2019), which yield a sensitivity of $0.42^{+0.31}_{-0.09}$ m/century/K. The balance temperatures corresponding to all TSLS estimates are listed in Table 1.

## 6 Discussion

We find that both model projections and observations show a near linear relationship between century averaged temperature change and the average rate of sea level rise (Figure 2). A linearization captures the bulk of the sea level response on these time scales. This shows that the concept is sound and that TSLS is a suitable new metric for assessing the graveness of global mean sea level changes.

The relationship deduced from model projections differs systematically from extrapolation of the observational relationship (Table 1 and Figure 2). Sea level projections assessed in AR5 have a substantially smaller TSLS than exhibited by historical observations, whereas SROCC is more comparable (Table 1). The greater SROCC sensitivity is driven by the warmest scenario and the higher TSLS is accompanied by a warmer balance temperature that is far from the observationally based estimate (Table 1). Future TSLS may well be different from the past due to non-linearities or non-stationarities in the relationship (Church et al., 2013). Thus, the discrepancy highlighted by Figure 2 does not necessarily demonstrate a bias in model projections, but as a minimum call for a yet to be prepared detailed explanation. Ideally, we would test the models using hindcasts to verify their ability to reproduce the past. Unfortunately, such hindcasts are unavailable for sea level projection models assessed in both AR5 and SROCC. This is critical as Slangen et al. (2017) identified substantial biases in hindcasts of Greenland surface mass balance, glacier mass loss, and deep ocean heating. Adjusting for these systematic biases increase the modelled sea level rise over the 20th century by ~50%. The discrepancy between historical and projected sensitivities is puzzling considering the lack of possibilities for a validation of the model projections.

In order for non-linearities to explain the discrepancy between the past and future relationship between warming and the rate of sea level rise, it is evident from Figure 2 that these would have to be sub-linear. This is incompatible with our current understanding. Major non-linearities are not expected this century according to the process knowledge encoded in the model projections assessed in both AR5 and SROCC, with SROCC presenting some signs of a super linear response (Figure 2). Antarctica, in particular, may have a super-linear response (Oppenheimer et al. 2019; DeConto and Pollard, 2016; Edwards et al. 2019; Bamber et al. 2019). Further, expert elicitation results overlap with the relationship found for the historical period but with a higher sensitivity (Table 1), which may be due to an anticipated super-linear response not captured by AR5 and SROCC assessment of model results. Antarctic rapid ice dynamics was considered as scenario independent in the IPCC fifth AR5 (Church et al., 2013), in stark contrast to later results (Oppenheimer et al. 2019; DeConto and Pollard, 2016; Edwards et al. 2019). We therefore propose AR5 to have a TSLS likely upper bound, which is biased low.

## 7 Conclusion

We define a new Transient Sea Level Sensitivity (TSLS) metric, which relates the rate of global mean sea level rise to global century-long mean surface temperature change. We find that this metric can account for most of sea level response to

temperature increase on this time scale. The TSLS metric is useful as it allows for model sensitivity comparisons, even if the models have not been run for the same set of scenarios, e.g. different radiative forcing. By framing the transient sensitivity in terms of temperature we separate the sea level sensitivity from climate sensitivity to a large extent. This allows for easier comparison between sea level models that are forced by different Earth system models. We propose that TSLS estimated from hindcast simulations can serve as a valuable emergent constraint of sea level models, although this is currently hampered by the lack of information needed to construct these.

We compare the model projections over the 21st century assessed by the IPCC with historical records from 1850-2017. We find that the model projections assessed in both AR5 and SROCC fall substantially below an extrapolation of historical records (Figure 2). This is reflected in the estimates of TSLS and balance temperature, which does not match the historical estimate (Table 1). Future sensitivity may be different from the past as the relationship between warming and sea level rate may be non-linear or non-stationary. We reason that a non-linearity cannot explain the mismatch as the required curvature would be inconsistent with process knowledge encoded by model projections assessed in SROCC and expert expectations (Oppenheimer et al. 2019; Bamber et al., 2019). Based on our analyses we cannot fully reject that the sensitivity between the historical period (1850-2017) and the projection period (2000-2100) differs. The major sea level contributors have characteristic response times of several centuries (Clark et al. 2018; Li et al., 2013; DeConto and Pollard, 2016; Oppenheimer et al. 2019; Church et al. 2013), which suggests that the sensitivity is unlikely to change substantially between these periods. The outcome of an expert elicitation is more consistent with an extrapolation of the historical relationship than AR5 and SROCC (Figure 2 and Table 1). Further, Slangen et al. (2017) identified substantial biases in process model hindcasts, which draws into question whether the AR5 and SROCC assessed models would be able to reproduce the time evolution of historical sea level rise. This is supported by our interpretation of the TSLS discrepancy between past and future. Our analysis implies that the model states used for the assessment in SROCC are too close to balance for present-day conditions and at the same time underestimate TSLS. Taken together this suggests that the projected global sea level rise by the end of this century in various IPCC reports are at best conservative and consequently underestimate the upper bound of what is referred to as the likely sea level rise by the end of this century.

**Data availability**

All data has previously been published and are publicly available.

**Author contributions**

AG designed the research study and conducted the analysis. AG and JHC interpreted the results and wrote the manuscript.

## Competing interest

The authors declare that they have no conflict of interest.

## Acknowledgements

Aslak Grinsted received funding from Villum experiment OldNoble grant number 28024, and Villum Investigator Project IceFlow grant number 16572. This work also received support by the European Union under the Horizon 2020 Grant Agreement 776613, the EUCP project.

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

**Table 1: Transient sea level sensitivity, and balance temperatures estimated from different sources. Intervals are likely ranges (17-83%). Symbols indicate that the difference from the observational estimate is significant at p<0.05 (\*), and p<0.1 (†) using a two-tailed test assuming normality.**

|  | Sea level sensitivity<br>m/century/K | Balance Temperature<br>°C |
| --- | --- | --- |
| Observations | 0.40 [0.35 – 0.44] | -0.70 [-0.77 – -0.64] |
| SROCC | 0.39 [0.36 – 0.43] | -0.14[†] [-0.42 – 0.23] |
| AR5 | 0.27\* [0.26 – 0.30] | -0.63 [-0.70 – -0.41] |
| Expert elicitation | 0.47 [0.33 – 0.85] | -0.37\* [-0.36 – -0.05] |

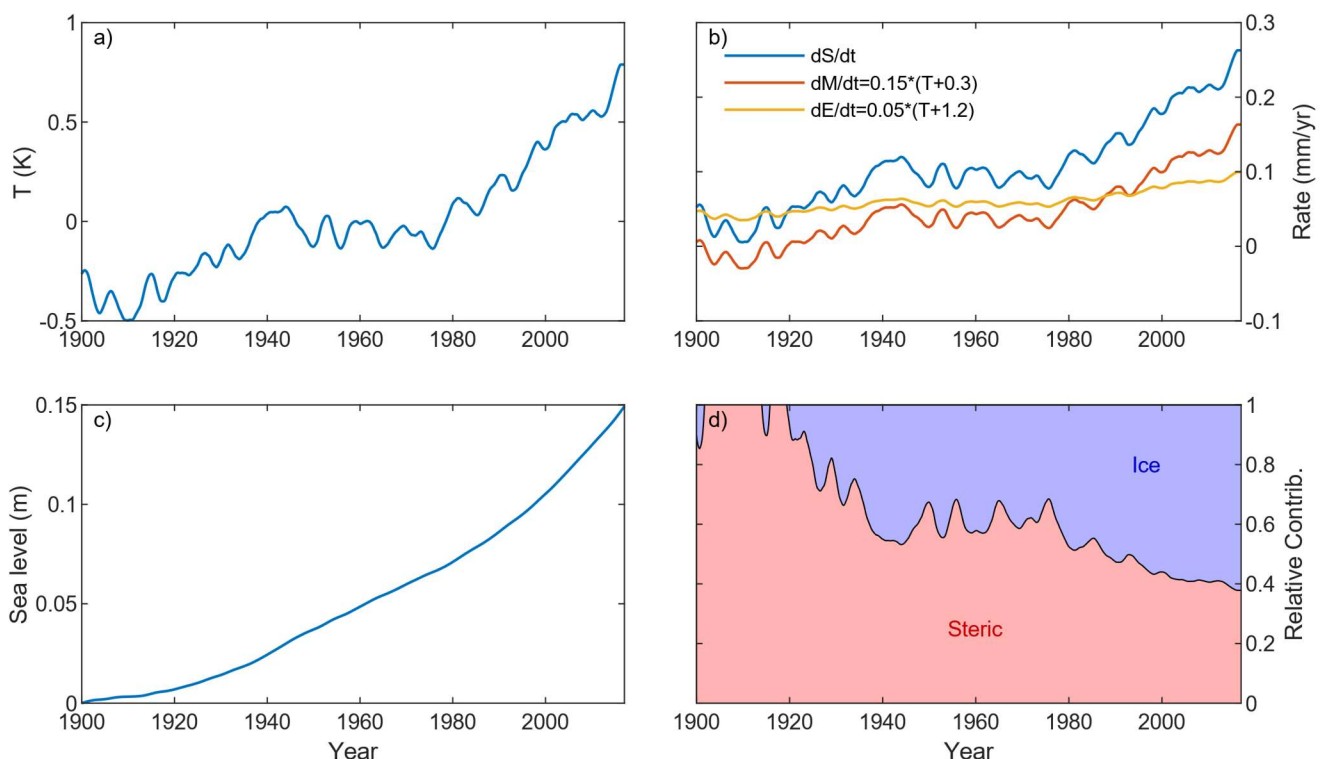

**Figure 1: Illustrative example demonstrating how changing relative sea level contributions can arise in a world where all contributors respond linearly to temperature. a) temperature forcing; b) The rate of sea level rise ($\dot{S}$) is modelled as the sum of two contributors: ice melt ($\dot{M}$) and steric expansion ($\dot{E}$); both contributions are modelled as linear in T. c) The sea level curve obtained by integrating $\dot{S}$. d) The relative contributions from ice melt and steric expansion (e.g. $\dot{E}/\dot{S}$).**

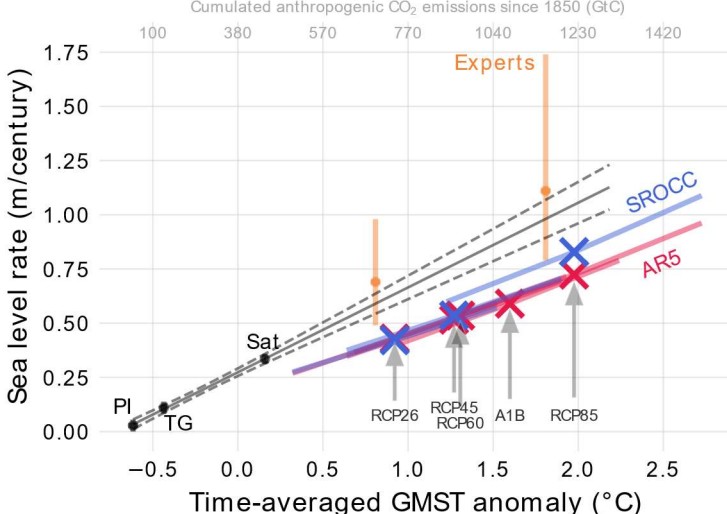

**Figure 2: The rate of sea level rise versus long term average temperature as seen in observations (black), in model projections (red/blue), and expectations in an expert elicitation (orange). Each point represents an average over a time period (PI: 1850-1900; TG: 1900-1990; SAT: 1993-2017; AR5/SROCC/Experts: 2000-2100). Sea level projections as assessed in AR5 and SROCC systematically fall below what would be expected from extrapolating observations to warmer conditions, as well as below the expert elicitation. Error bars show estimated likely ranges (17-83%). Likely ranges for SROCC and AR5 are shown as slanted error bars.**

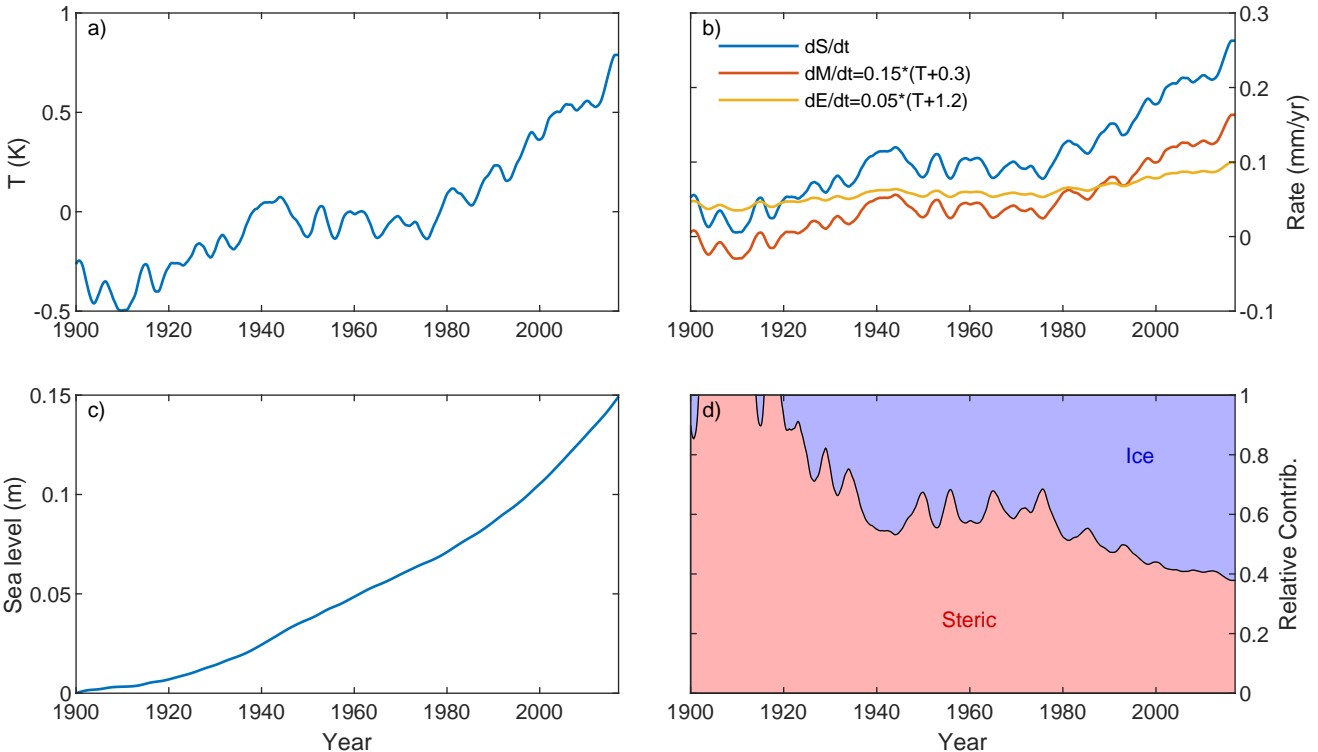

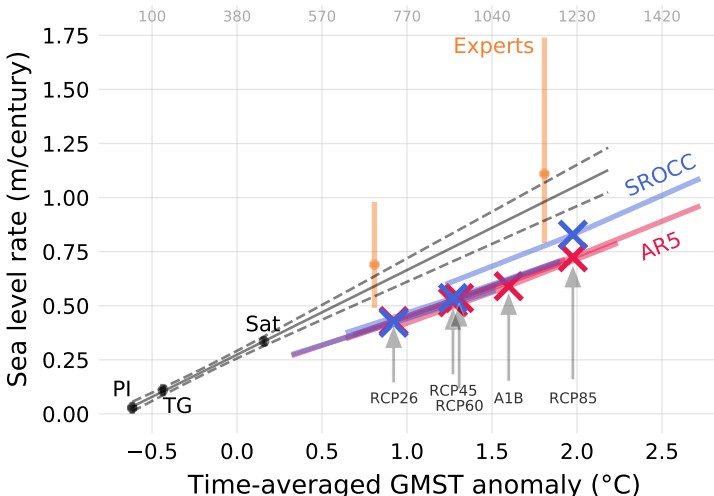