# Peer review of "The transient sensitivity of sea level rise"

_Ocean Science, 2020_

## Referee Comment (RC1) · Tal Ezer (Referee) · 5 Aug 2020

General Comments: The (very short) paper looks at linear relations between global sea level rise (SLR) rates and time-mean temperatures in both observations and climate model projections- the results suggest that models may underestimate future sea level rise, which is a very important finding. The study is clearly written, and the results are interesting, though since I am not a global climate modeler, I am not sure if this result about the SLR-SST relation in models is new or already known to climate modelers. There are several caveats in the study with its very condensed presentation (only one figure and 1 table), that are needed to be explained (with potentially expanded calculations).

[Figure]

Major Comments: There are several assumptions that are not completely correct, so their impact should be addressed more extensively.

1. SLR rates are far from being linear, they are in general accelerating, but there are also significant multi-decadal variations in SLR rates (e.g., see Frederikse et al., Nature, 2020, doi:10.1038/s41586-020-2591-3). Therefore, the assumption that the SLR-SST linear relation in the past should be the same as in the future may not hold. Moreover, the period chosen for time-averaged SST and SLR may affect the results-some experiments to see how sensitive the results are to different chosen periods may be useful.

2. The SLR-SST relation assumes that SLR is related to SST through thermal expansion, but what about the contribution from water masses? In recent years and in the future contribution to SLR from ice melt will increase relative to thermal expansion (Frederikse et al. 2020, and many others). This by itself may explain the main results here. To see if this is the case, you may add to the calculation results from the same models over the same period as the observations to see if the results are due to model biases or the neglection of water mass contribution.

3. Linear regression in Fig. 1 is obtained from only ∼5 points, can accuracy be improved by regression over several models, not just the mean of each scenario? Are there for example, models (recent high-resolution) that do follow the observed line? These suggestions may be outside the scope of the study but would greatly help to explain the results and its implications.

Minor Comments: 4. Lines 9-10: "To understand this discrepancy"- I am not sure this is a real discrepancy or just different estimations of future changes.

5. Line 38: "... century averaged temperature"- can you define exactly over what period the averaged was calculated (in Fig. 1 it says $CO_2$ since 1850). As mentioned before, it will be useful to know how sensitive the results are to the chosen period, given the non-linear nature of SST and SLR.

6. In Fig. 1, what are the superscript numbers above labels (numbered references left from a previous submission?)

7. In Table 1, only 1 out of 4 sensitivity numbers is statistically significant... can this be improved by larger set of data from different models, as suggested above? Is there physical meaning to the "balance temperature"?
* * *

---

## Referee Comment (RC2) · Anonymous Referee #2 · 12 Aug 2020

Summary: The manuscripts defined a transient sea-level rise sensitivity as the linear dependency of the rate of sea-level with centennially averaged global mean temperature (surface? ) temperature. The authors estimate this sensitivity from observations and from future climate simulations from the CMIP5 model ensemble. They conclude that the model-derived values are smaller than those derived from 'observations' and thus the future sea-level rise may become larger than those projected by climate models.

Recommendation: This is a surprisingly short manuscript, which in my view leaves many technical detailed unclear. It does not have a result section, and so it was for me difficult to interpret what the sole figure 1 and the sole table 1 is actually representing. The very concept of transient sea-level sensitivity requires a much deeper physical

discussion, My impression is, therefore not positive,. The manuscript seems in many respects to be incomplete.

1) The definition of sea-level climate sensitivity, although used in some previous studies, is at least rather questionable, and it was clearly questioned also in the AR5 report itself. This manuscript should at the very least justify in the first place why this concept is meaningful. For instance global mean sea-level rise is brought about by two very different mechanisms: expansion of the water column and melting of land ice. A back-of-the-envelope calculation yields that the global sea-level rise caused by the capture of an energy flux of 1 w/m2 by the liquid ocean, and its subsequent expansion, is about 1.9 mm. This is very different from the sea-level rise caused by the capture of of 1 w/m2 by land-ice and subsequent melting, assuming the ice is already at 0C, (94mm). Of course, this also depends on where the heat flux is captured and many regional details, but the difference between 1.9mm and 94mm is in principle enormous. Therefore, the very concept of a linear relationship between energy flux imbalance and the rate of global sea-level rise is physically questionable, at least it requires a plausible justification, as the 'sensitivity' depends on the relative contribution of thermal expansion and melting. This contribution is rather uncertain for the future, but it seems to me clear that in the near future melting will play a much bigger role through glacier melting, then perhaps a smaller role as glaciers are completely melted and then again a bigger role when melting in Greenland and Antarctica sets in. So it is really difficult for me to envisage a simple linear relationship to describe this dependency. It may be that in practice it works, but this needs to be justified. Unfortunately, I do not see which data could be used to justify this assumption. The centennial smoothing assumed in this study would require several millennia of data for a robust justification.

2) Related to point 1, the CMIP5 global climate models do not include land ice melting. This is the reason why the IPCC AR5 included a contribution to estimated sea-level rise by expert knowledge. But I wonder how the comparison between AR5 models and observations can be meaningful, when one of the key components is missing in
the models. Therefore, it is not really surprising that the sensitivity estimated from models is smaller than that estimated from observations. This is again the reason why the IPCC augmented the estimated sea-level rise by 2100 with an approximate contribution from land-ice melting.

3) The approach in this manuscript seems rather similar to the approach by Rahmstorf (2007). The reader would like to know in what aspects both approaches differ, and how this difference my affect the results.

4) I struggle to understand what Figure 1 and Table 1 are exactly showing ? Certainly the caption or the main text should include a much lengthier description. Points that remain unclear to me are: what is the averaging window (100 years as suggested in the main text?) If yes, the global mean temperature observations would be just 1 point ?), What does the point labeled as Sat9 represents ? Probably it represents the data in the satellite era, but there is no mention of this in the main text, only one paper listed in the reference list. The same can be said about TG7. To be honest, at this point I wonder whether the authors have carefully checked the manuscript before submitting.

In the case of observations, if my interpretation is correct, the linear fit is constructed using two points, both with different characteristics(one represents centennial means, the other satellite-era means). Is linear fit with just two points enough to be extrapolated ? The extrapolation would be even more questionable when considering that the physical processes would change over time, as explained in my point 1. How were the uncertainties calculated considering that the errors in each of these data points are different ?

Further points

5) The main text mentions reconstructions of sea-level in the preindustrial period, but were have they been used ? There is no mention of temperature reconstructions that could be used for the estimation of sea-level sensitivity

6) The caption of the table mentions a level of significance in the difference of the sea-level sensitivity. How has it been calculated ?

7) The temperature anomaly are referred to the base line 1986-2005. What is the reason for this short base line, when the link between T and sea-level rate is assumed to be at centennial scales ? It does not seem consistent. I guess there is an explanation for it, but the manuscript is so short and concise that the reader is left wondering

The latter are just examples of open technical questions that should be clear in a properly formatted manuscript, with proper length

---

## Referee Comment (RC3) · Anonymous Referee #3 · 13 Aug 2020

In this manuscript, the authors define the new concept of transient sea level sensitivity that is inspired by the transient climate sensitivity but that is adapted to the sea level problem. In particular it relates the sea level rise over a century with the average temperature anomaly compared to a steady state over the same period.

I think this concept, even with all its drawbacks, has the potential to be useful but the arguments developed in this manuscript needs to be further developed to be convincing. Especially since the authors make important claims about the underestimation of future sea level rise by the IPCC AR5 and SROCC process-based method.

General comments:

An important motivation to define the TSLS is the linear relationship between sea level

change and GMST in both observations and models. However that relationship is not very convincing. I agree with the theoretical points mentioned by referee #2 so I will not come back on those but I will focus on the observations and model data used in Figure 1:

1) The observational data used here to back up such a relationship is weak. There are only three points, moreover the pre-industrial and tide gauge periods are very close to each other. With therefore the main point driving the slope of the linear relation being the satellite period which is only around 25 years. I would suggest that if the author think 25 years is enough to estimate the TSLS then the tide gauge period could be split in a few 25 years periods.

2) For model data the uncertainty lines are obtained from the assumption of full covariance between GMST and sea level uncertainties in IPCC projections. But that is not the case at all, there are many sources of uncertainty in the sea level projection that are independent of temperature. For example Greenland and Antarctic ice dynamic contribution, glacier model uncertainty (four different models are used in AR5 and SROCC). The assumption is justified by the fact that when it is made it shows a linear relationship between GMST and sea level but this is what the authors try to demonstrate. Also for SROCC the linearity doesn't seem to hold at all.

l.47: "This does not automatically demonstrate a bias in model projections, but as a minimum call for a detailed explanation."

Since this is the main claim of this short paper I think attempting to provide an explanation falls on the shoulders of the authors. There is already some literature on that subject see for example Slangen et al. 2017, in particular section 4:

"When all the contributions are combined, the models add up to a GMSL change of 92 6 47mm for the period from 1901–20 to 1996–2015 (Table 4, Fig. 9a). Compared to the average of the four reconstructed global mean time series for the overlapping period from 1901–20 to 1988–2007 (Table 5, Fig. 9a, the model simulations clearly underestimate the observed GMSL and explain only 50% 6 30% of the observed change (using 61.65s of the models to the mean of the observations)."

And the following discussion on adding corrections to the sea level computed from the models to solve the issue.

Small comments:

- Figure 1: I can't find an explanation for the numbers in PI11, TG7, Sat9 and others.

Slangen, Aimée B. A., Benoit Meyssignac, Cecile Agosta, Nicolas Champollion, John A. Church, Xavier Fettweis, Stefan R. M. Ligtenberg, et al. "Evaluating Model Simulations of Twentieth-Century Sea Level Rise. Part I: Global Mean Sea Level Change." Journal of Climate 30, no. 21 (November 2017): 8539–63. https://doi.org/10.1175/JCLI-D-17-0110.1.

---

## Referee Comment (RC4) · Anonymous Referee #4 · 22 Aug 2020

The paper The transient sensitivity of sea level rise by Grinsted and Christensen discusses the relationship between global mean surface temperature and global mean sea level rise on a time scale of the order of a century. The authors acknowledge earlier work on the topic and frame the relation between temperature and sea level rise as an independent proxy for the evaluation of recent assessments of sea level rise projections that are biased low compared to observations. The article claims a linear sea level sensitivity of 0.4 m/century/K based on observations and either lower sensitivity in AR5 or higher balance temperature in SROCC and Bamber et al., 2019., respectively.

**General comments**

The paper is very short and concentrates on the discussion of the discrepancy between the parameters of linear regressions between averaged global mean surface

temperature and global mean sea level rise, based on observations (past) and climate projections (future). In the face of high and rising stakes on the response to sea level rise additional proxies for the evaluation of projections of sea level rise are needed. The paper contributes to this end in bringing back the sea level sensitivity into the discussion. I think it is worth to be published and discussed in the community. The paper misses the opportunity to go deeper into the matter and offer thoughts or strategies how to address the discrepancies in transient sea level sensitivity between observations and climate projections.

**Specific comments**

I wonder whether we could learn something more about the impact of model development if the current analysis would include older projections like AR3 and AR4. Those were already below GMSL rise according to Rahmstorf 2007, Horton et al. 2008.

One weak point of the analysis, as I see it, is the different ranges of GMST used for the regressions of the observations and model projections. Would it be possible and useful to include model estimates from paleo runs that had GMST anomalies in the same range as those projected for the 21st century?

The regression lines in Fig. 1 should pass through the mean time-averaged GMST anomaly and the mean sea level rate. Is there any information contained in the scatter of the mean GMST and mean GMSL rate of the individual regressions?

It would be interesting to discuss some of the physical processes, thresholds, time scales and limitations, that would render the relationship between averaged GMST and GMSL rate non-linear. It would help to establish the transient sea level sensitivity as a metric next to equilibrium sea level rise on longer time scales.

Are current climate models or model ensembles good enough so that their uncertainty in GMST was smaller than the uncertainty in balance temperature in Table 1? Is the spread in balance temperature inherent in climate models or does it come from the

combination of climate models (GMST, steric) with process models (ice sheets dynamics)?

From Table 1 one could deduce sea level rise of 0.28, 0.05, 0.17 and 0.17 m/century at balance temperature. The 0.28 m/century sea level rise in the observations at balance temperature is already above the 0.1-0.2 m/century sea level rise for the 20th century. Since sea level rise is accelerating we are probably above balance temperature since at least the satellite era. This seems to point to a contradiction in the data and the assumption of a linear process. How can the balance temperature be interpreted or how well can we know it?

**Technical corrections**

I6: assessments from the Intergovernmental Panel on Climate Change implies I20: and melts.. A perturbation I20: perturbation in greenhouse gas concentrations change I47: table 1 and figure 1 I52: table 1 I63: figure 1 I69: figure 1

СЗ

---

## Referee Comment (RC5) · Anonymous Referee #4 · 2 Sep 2020

Sorry, for not clearly stating what I meant under technical corrections. In Line 6 (and vice versa in Line 20) the subject is third person plural but the verb is conjugated as third person singular. In Line 20 there are two dots after 'melts'. The other comments in Lines 47, 52, 63 and 69 are about the capitalization of names like Table 1.

- l6: assessments from the Intergovernmental Panel on Climate Change implies - l20: and melts.. A perturbation - l20: perturbation in greenhouse gas concentrations change - l47: table 1 and figure 1 - l52: table 1 - l63: figure 1 - l69: figure 1

---

## Author Comment (AC5) · 2 Sep 2020

This document outlines our plan for the revision. It is gathered from our responses to the four review comments and grouped by topic.

**Limitations of the metric**

- Elaborate substantially on the limitations of the metric.
- Emphasize more strongly the limitations of the comparison to the observational estimate. Especially in abstract.
- Expand the discussion and emphasize more strongly the limitations of the comparison to the observational estimate.

- Stress even more limitations of a comparison between two different periods: historical and projections.
- Talk more about the limitations of the TSLS.

**Future may be different from past**

- Discuss the physical mechanisms behind the relationship and thereby stress that sensitivity may be different in future from past, and that this could potentially explain "the discrepancy".
- Ensure careful phrasing of the conclusions. We do not want to overstate the significance of "the discrepancy" between past and future. But we will emphasize the caveats related to the use of GCM climate projections further processed to get SLR information.
- Stress that sensitivity may be different in future from past, and that this can possibly explain "the discrepancy" and assess the involved physical mechanisms more clearly.
- Stress that we do not expect TSLS to be constant over time.

**Extrapolation is a comparison, not projection**

- Expand the discussion to better clarify that extrapolation is only used for a comparison, and not a projection.
- Stress that extrapolation is not a projection but plotted for comparison.

**Clarity on time periods**

OSD
- Discuss time periods more clearly. Both in figure caption, and when introducing TSLS.
- State time-intervals in figure caption.
- Discuss "century time scale" choice more in main text.
- Explain more in main text time period.

**Figure**

- Remove superscripts from figure.
- Expand description of figure.
- Expand caption Explain what each point is, especially their time span.

**Statistics**

- Explain statistical methods in detail.
- Add a more complete description that explains that full covariance is unlikely, and how it impacts results.

**Miscellaneous**

- · Check if it makes sense to move GMST definition into the main body of text.
- Point out that AR5 SROCC have no hind casts in their presentation of the SLR discussions and it has therefore not been demonstrated that these models can reproduce past sea level rise.
- Address explicitly the premises adopted in AR5 (and implicitly in SROCC) that a universal linear relationship between sea level rise rate and temperature is questionable.
- Discuss non-linearity and non-stationary.
- Consider discussing common misconception.
- Call for hindcast validations for future sea level projections.
- Call for historical validation of models used for sea level projections. Not just of the individual contributor models, but also of the aggregate model.
- Be explicit about baseline motivation.
- Explain that we only use published estimates, and motivation.
- Discuss Slangen2017 as context.
- · Add more to the motivation part of the manuscript
- Add an outlook for how TSLS discrepancies can be addressed.
  - Brainstorm to consider when revising:
  - Ensure that projection models also have hindcasts of the historical past.
  - Look into the transient sensitivity of individual contributors.
  - Understand how TSLS changes over time
  - Model studies to understand the limitations of TSLS.
- Consider adding a short speculative paragraph on uncertainties in balance temperature.

---

## Author Comment (AC6) · 14 Sep 2020

Thank you. These will be adressed and checked by a native english speaker.
* * *

---

## Author Response (AR1)

**Dear Editor**

The reviewers raised many good questions and concerns, but none raised any critical issues with our analysis. The most serious concerns were centered around that we were not sufficiently explicit with regards to the limitations of our new TSLS metric. Secondly, it has become clear that the very brief letter structure of the original manuscript was not ideal. The reviewers wanted a more expanded discussion of the manuscript. We have restructured the manuscript to follow a more traditional paper outline (intro-data-methods-results-discussion-conclusion).

In the replies to the reviewers we stated how we planned to address each of their concerns. The individual points of the revision plan is collated below, followed by a description of *the revisions in red italics*.

Best regards,

Aslak Grinsted and Jens Hesselbjerg Christensen

**Description of revisions**

**Revision plan and revisions:**
**Limitations of the metric**

- Elaborate substantially on the limitations of the metric.
- Emphasize more strongly the limitations of the comparison to the observational estimate. Especially in abstract.
- Expand the discussion and emphasize more strongly the limitations of the comparison to the observational estimate.
- Stress even more limitations of a comparison between two different periods: historical and projections.
- Talk more about the limitations of the TSLS.

*We have expanded considerably on all these points in the introduction, discussion, and in the conclusion. The revised text is much more explicit on the limitations of the metric. Here are some examples of some revisions:*

*Introduction: "… However, there may be processes that can cause future sensitivity to be different from the past (Church et al., 2013). These changes can broadly be categorized as being due to a non-linear response to forcing, or due to a non-stationary response where the response depends on state of the system. E.g. the sensitivity of small glaciers to warming will depend on how much glacier mass there is left to be lost, and we therefore expect this to have a non-stationary response. Nature is complex and will be both non-linear and non-stationary, and this places limits on extrapolation."*

*Discussion: "Future TSLS may well be different from the past due to non-linearities or non-stationarities in the relationship (Church et al., 2013). …"*

*Conclusion: "Future sensitivity may be different from the past as the relationship between warming and sea level rate may be non-linear or non-stationary. We reason that a non-linearity cannot explain the mismatch as the required curvature would be inconsistent with process knowledge encoded by model projections assessed in SROCC and expert expectations (Oppenheimer et al. 2019; Bamber et al., 2019). Based on our analyses we cannot fully reject that the sensitivity has changed between the historical period (1850-2017) and the projection period (2000-2100)."*

**Future may be different from past**

- Discuss the physical mechanisms behind the relationship and thereby stress that sensitivity may be different in future from past, and that this could potentially explain "the discrepancy".
- Ensure careful phrasing of the conclusions. We do not want to overstate the significance of "the discrepancy" between past and future. But we will emphasize the caveats related to the use of GCM climate projections further processed to get SLR information.
- Stress that sensitivity may be different in future from past, and that this can possibly explain "the discrepancy" and assess the involved physical mechanisms more clearly.
- Stress that we do not expect TSLS to be constant over time.

*We have expanded considerably on past vs future which is closely linked to the limitations of the metric. The revised text is much more explicit on the limitations of the metric. Here are some examples of some revisions:*

*Introduction: "… However, there may be processes that can cause future sensitivity to be different from the past (Church et al., 2013). These changes can broadly be categorized as being due to a non-linear response to forcing, or due to a non-stationary response where the response depends on state of the system. E.g. the sensitivity of small glaciers to warming will depend on how much glacier mass there is left to be lost, and we therefore expect this to have a non-stationary response. Nature is complex and will be both non-linear and non-stationary, and this places limits on extrapolation."*

*Discussion: "Future TSLS may well be different from the past due to non-linearities or non-stationarities in the relationship (Church et al., 2013). …"*

*Conclusion: "Future sensitivity may be different from the past as the relationship between warming and sea level rate may be non-linear or non-stationary. We reason that a non-linearity cannot explain the mismatch as the required curvature would be inconsistent with process knowledge encoded by model projections assessed in SROCC and expert expectations (Oppenheimer et al. 2019; Bamber et al., 2019). Based on our analyses we cannot fully reject that the sensitivity has changed between the historical period (1850-2017) and the projection period (2000-2100)."*

**Extrapolation is a comparison, not projection**

- Expand the discussion to better clarify that extrapolation is only used for a comparison, and not a projection.
- Stress that extrapolation is not a projection but plotted for comparison.

*We now explicitly and systematically refer to the extrapolation of the historical relationship as an extrapolation. The projection vs extrapolation is discussed in the introduction, and in general where we discuss the limitations of the TSLS metric.*

**Clarity on time periods**

- Discuss time periods more clearly. Both in figure caption, and when introducing TSLS.
- State time-intervals in figure caption.
- Discuss "century time scale" choice more in main text.
- Explain more in main text time period.

*Done! In the previous manuscript the periods used where mainly written in the data & methods section which was almost an appendix. We have restructured the manuscript so that the periods used are much more clear in data section, discussion, and conclusion and in the captions.*

*The "century time scale" is discussed in the context of non-stationarity, limitations of the metric, and past is not future.*

**Figure**

- Remove superscripts from figure.
- Expand description of figure.
- Expand caption – Explain what each point is, especially their time span.

*Done!*

**Statistics**

- Explain statistical methods in detail.
- Add a more complete description that explains that full covariance is unlikely, and how it impacts results.

*Done! We have restructured the manuscript so that it now has a separate methods section.*

**Miscellaneous**

- Check if it makes sense to move GMST definition into the main body of text.

*Moved.*

- Point out that AR5 & SROCC have no hind casts in their presentation of the SLR discussions and it has therefore not been demonstrated that these models can reproduce past sea level rise.

*Done!*

- Address explicitly the premises adopted in AR5 (and implicitly in SROCC) that a universal linear relationship between sea level rise rate and temperature is questionable.

*Done! (see also past is not future and limitations of the metric)*

- Discuss non-linearity and non-stationary.

*Done. New text in intro: "However, there may be processes that can cause future sensitivity to be different from the past (Church et al., 2013). These changes can broadly be categorized as being due to a non-linear response to forcing, or due to a non-stationary response where the response depends*

*on state of the system. E.g. the sensitivity of small glaciers to warming will depend on how much glacier mass there is left to be lost, and we therefore expect this to have a non-stationary response."*

- Consider discussing common misconception.

*Distraction. Not included.*

- Call for hindcast validations for future sea level projections.
- Call for historical validation of models used for sea level projections. Not just of the individual contributor models, but also of the aggregate model.

*We now discuss hind-casts in both the discussion and conclusion. It should be clear from context that we consider hind casts are very desirable (even ideal), but we do not explicitly call for them. We write: "Ideally, we would test the models using hind casts to verify their ability to reproduce the past. Unfortunately, such hind-casts are unavailable for sea level projection models assessed in both AR5 and SROCC. This is critical as Slangen et al. (2017) identified substantial biases in hind-casts of Greenland surface mass balance, glacier mass loss, and deep ocean heating. These biases increase the modelled sea level rise over the 20th century by ~50%."*

- Be explicit about baseline motivation.

*Done!*

- Explain that we only use published estimates, and motivation.

*Done!*

- Discuss Slangen2017 as context.

*Done! (See discussion and conclusion and notes about hind-casts)*

- Add more to the motivation part of the manuscript

*The introduction has been expanded.*

- Add an outlook for how TSLS discrepancies can be addressed.
  - Brainstorm to consider when revising:
  - Ensure that projection models also have hindcasts of the historical past.
  - Look into the transient sensitivity of individual contributors.
  - Understand how TSLS changes over time
  - Model studies to understand the limitations of TSLS.

*We have not added an outlook as the most important point is highlighted in the manuscript elsewhere. "Ideally, we would test the models using hind casts to verify their ability to reproduce the past. Unfortunately, such hind-casts are unavailable for sea level projection models assessed in both AR5 and SROCC. This is critical as Slangen et al. (2017) identified substantial biases in hind-casts of Greenland surface mass balance, glacier mass loss, and deep ocean heating. These biases increase the modelled sea level rise over the 20th century by ~50%."*

- Consider adding a short speculative paragraph on uncertainties in balance temperature.

*We have not added this. A distraction and possibly too speculative.*

**Response to RC1: Tal Ezer**

*General Comments:*

*The (very short) paper looks at linear relations between global sea level rise (SLR) rates and time-mean temperatures in both observations and climate model projections- the results suggest that models may underestimate future sea level rise, which is a very important finding. The study is clearly written, and the results are interesting, though since I am not a global climate modeler, I am not sure if this result about the SLR-SST relation in models is new or already known to climate modelers. There are several caveats in the study with its very condensed presentation (only one figure and 1 table), that are needed to be explained (with potentially expanded calculations).*

It was our intent to write a brief discussion letter where we introduce the Transient Sea Level Sensitivity metric. We hope this metric will be adopted by the community as a simple way to compare the first order transient response between different models. In the paper we plot the results of published work in a thought-provoking way. It is therefore our opinion that this is much better suited as a discussion letter, rather than a longer more traditional article.

We are not the first to note that there must be some relationship between sea level rate and temperature. Awareness of this is evident already in the first IPCC assessment report, and the idea was explicitly exploited by Rahmstorf (2007) to construct a semi-empirical model projection. So, we do not consider this to be the main contribution of the paper, although clearly many modelers are not aware of these developments. In our opinion the main contributions of our paper are more prominent and here emphasized explicitly in order of importance:

1) The introduction of the Transient Sea Level Sensitivity (TSLS) metric.
2) The finding that a straight line is a good approximation to the transient response in the models assessed in AR5 and SROCC. I.e. that TSLS is a useful metric that captures most of the transient response according to present physical understanding.
3) The highlighted apparent discrepancy between the TSLS of models used for sea level projections and historical data.

Note especially, that we do not consider the observational extrapolation to be a projection. We explicitly say: "*This does not automatically demonstrate a bias in model projections, but as a minimum call for a detailed explanation*". This is intended as a very clear and explicit caveat, and a call for further work. So, we agree with the referee that more analysis is needed to understand the discrepancy, but also that this is beyond the scope of this discussion letter. However, we gather from the full set of reviews that we need to be more explicit about the limitations of the TSLS metric and will make this point more prominent.

**Revision plan:**

- Elaborate substantially on the limitations of the metric.
- Discuss the physical mechanisms behind the relationship and thereby stress that sensitivity may be different in future from past, and that this could potentially explain "the discrepancy".
- Expand the discussion to better clarify that extrapolation is only used for a comparison, and not a projection.

*Major Comments: There are several assumptions that are not completely correct, so their impact should be addressed more extensively.*

*1. SLR rates are far from being linear, they are in general accelerating, but there are also significant multi-decadal variations in SLR rates (e.g., see Frederikse et al., Nature, 2020, doi:10.1038/s41586-020-2591-3). Therefore, the assumption that the SLR-SST linear relation in the past should be the same as in the future may not hold. Moreover, the period chosen for time-averaged SST and SLR may affect the results some experiments to see how sensitive the results are to different chosen periods may be useful.*

We do not assume "*that the SLR-SST linear relation in the past should be the same as in the future*". We simply compare past with future sensitivity and note that there is a discrepancy. But we also stress that "*This does not automatically demonstrate a bias in model projections, but as a minimum call for a detailed explanation*" and "*Future TSLS may well be different from the past…*". We do this for exactly that reason – we will emphasize this even further.

We do not assume a steady acceleration over time. There is multi decadal variability temperature, and that should be reflected in the sea level rate. We get the reviewers point though: that the simple straight line cannot capture all variability. We acknowledge that the TSLS metric is a simplification of a complex system. It can only characterize the first order response. But this is no different from established metrics such as the Transient Climate Response which have proven their usefulness. We will emphasize that this is exactly how we see the value of studying TSLS.

**Revision plan:**

- Stress that sensitivity may be different in future from past, and that this can possibly explain "the discrepancy" and assess the involved physical mechanisms more clearly..Discuss time periods more clearly. Both in figure caption, and when introducing TSLS.

*2. The SLR-SST relation assumes that SLR is related to SST through thermal expansion, but what about the contribution from water masses? In recent years and in the future contribution to SLR from ice melt will increase relative to thermal expansion (Frederikse et al. 2020, and many others). This by itself may explain the main results here. To see if this is the case, you may add to the calculation results from the same models over the same period as the observations to see if the results are due to model biases or the neglection of water mass contribution.*

We agree that changes in the state of the climate system between the 20[th] and the 21[st] century, could potentially explain the discrepancy between the sensitivity in past and in the future. But without further analysis this is speculation. We write: "*This does not automatically demonstrate a bias in model projections, but as a minimum call for a detailed explanation*".

We also agree that it would be great if we could plot the results of the AR5&SROCC models for the historical period to compare to the historical data. Unfortunately, that is simply not possible because such historical runs were never made with the same aggregate model that was used for projections. This lack of a validation is precisely the reason why we feel that it is necessary to compare past and future sensitivity even if this may be an imperfect comparison.

The SLR-SST relation does not hinge on an assumption "*that SLR is related to SST through thermal expansion*". We do not assume this, and we do not neglect the ice mass contribution. Every point in figure 1 include both expansion and water mass contributions.

We do not assume that the relative proportions of the different sea level contributors remain
the same. Further, changing proportions is insufficient explanation of the discrepancy. We
illustrate this in Note R1 at the end of this document.

**Revision plan**

• Point out that AR5 & SROCC have no hind casts in their presentation of the SLR
discussions and it has therefore not been demonstrated that these models can
reproduce past sea level rise.

*3. Linear regression in Fig. 1 is obtained from only ~5 points, can accuracy be improved by*
*regression over several models, not just the mean of each scenario? Are there for example,*
*models (recent high-resolution) that do follow the observed line? These suggestions may be*
*outside the scope of the study but would greatly help to explain the results and its implications.*

The problem is that sea level rise is not an output from current generation Earth System Models
ESM. E.g. The contribution from Greenland is calculated by driving an ice sheet model and a
regional climate model with projected weather from an ESM. The total sea level rise is the sum
of the contribution from many processes – each with their own model. It is therefore
challenging to talk about a recent high-resolution model, as it is a combination of many different
models. This is what the IPCC provides, and they also attempt to account for modelling
uncertainties as well as possible. The likely range of the IPCC projections are presumably
intended to be a fair representation of the modelling uncertainty, and should therefore span
recent high resolution models.

A way to understand the discrepancy we observe would be to study how the IPCC models
reproduce the historical rates of sea level rise. If hindcasts can reproduce the PI, TG, and SAT
rates, then there is no issue. If not, then the historical sea level budget of the models can be
dissected to understand if there are issues. Unfortunately, this is not done in the IPCC reports, as
they only run the models used for projections for the 21st century and do not show hind casts.

**Revision plan:**

• Point out that AR5 & SROCC have no hind casts in their presentation of the SLR
discussions and it has therefore not been demonstrated that these models can
reproduce past sea level rise.
• Call for hindcast validations for future sea level projections.

*Minor Comments:*

*4. Lines 9-10: "To understand this discrepancy"- I am not sure this is a real discrepancy or just*
*different estimations of future changes.*

This is a question of wording. The difference is a discrepancy, even if there is an as of yet
unknown explanation for it.

**Revision plan:**

• Stress that sensitivity may be different in future from past, and that this could
potentially explain "the discrepancy".

- Emphasize more strongly the limitations of the comparison to the observational estimate. Especially in abstract.

*5. Line 38: "... century averaged temperature"- can you define exactly over what period the averaged was calculated (in Fig. 1 it says CO2 since 1850). As mentioned before, it will be useful to know how sensitive the results are to the chosen period, given the non-linear nature of SST and SLR.*

In the current version of the manuscript the time periods are mentioned in the methods section. In the plot the different points were calculated over different time intervals:

- SAT: 1993-2017
- TG: 1900-1990
- PI: 1850-1900
- AR5/SROCC/Experts: 2000-2100

At the moment this is very briefly mentioned in the methods section. Thus, the observational fit is based on data from 1850-2017, and projections are based on data from 2000-2100.

**Revision plan:**

- State time-intervals in figure caption.
- Discuss "century time scale" choice more in main text.

*6. In Fig. 1, what are the superscript numbers above labels (numbered references left from a previous submission?)*

That is correct. This will be fixed.

**Revision plan:**

- Remove superscripts from figure.
- Expand caption – Explain what each point is, especially their time span.

*7. In Table 1, only 1 out of 4 sensitivity numbers is statistically significant... can this be improved by larger set of data from different models, as suggested above?*

As mentioned above then there is no ensemble of models that we can draw from. Also, the aim of the IPCC assessments is to capture the full uncertainty. So presumably the AR5 and SROCC would span the distributions based on different models.

In our view the significance test is just a tool to help us avoid over-interpreting small differences between rows in the table. It is better that these tests are conservative, and we have therefore no goal of improving the significance. Indeed, it would be nice if the entire table was insignificant because that would mean that all the estimates were more consistent with the observational estimates.

There are four TSLS rows in table 1, and we test if they are significantly different from the first row (the observational estimate). So, there are only three tests for TSLS, not four. These shows:

- That expert estimates are not incompatible with historical data.

•     That the AR5 TSLS is significantly smaller than the historical TSLS.

•     That the SROCC TSLS (as estimated over the entire range) is in better agreement with historical data.

**Revision plan:**

•     Explain statistical tests in methods section.

|  *Is there physical meaning to the "balance temperature"?*

Yes. It can be framed as the amount of cooling needed to stop sea level rise (in the short term).

**Note R1: Changing proportions, yet constant sensitivity**

The sea level budget is changing, and we expect ice sheet melt to increasingly dominate the
budget. This might lead one to argue that the sensitivity must be changing as we don't expect
the individual contributors to be equally sensitive to warming. In this section we present a case
for why that is a flawed argument. We show that even in a completely linear model the relative
proportions of the individual sea level contributors can change.

Let's assume for the moment, that the rate of sea level rise is just the sum of the contribution
from ice melt ($\dot{M}$) and the contribution from thermal expansion ($\dot{E}$). We write:

$\dot{S} = \dot{M} + \dot{E}$

Let's also assume that these two contributions respond linearly to warming.

$\dot{M} = a_M T + b_M$

$\dot{E} = a_E T + b_E$

We insert and get a linear model for the sea level rate:

$\dot{S} = (a_M + a_E)T + b_M + b_E$

The proportion of sea level rise due to ice melt becomes

$\dfrac{\dot{M}}{\dot{S}} = \dfrac{a_M T + b_M}{(a_M + a_E)T + b_M + b_E}$ .

This is not generally constant in T. This demonstrates that a changing proportion of ice melt
does not necessarily imply a changing sensitivity to warming.

**Response to RC2: Anonymous referee #2**

*Summary: The manuscripts defined a transient sea-level rise sensitivity as the linear dependency of the rate of sea-level with centennially averaged global mean temperature (surface? ) temperature. The authors estimate this sensitivity from observations and from future climate simulations from the CMIP5 model ensemble. They conclude that the model-derived values are smaller than those derived from 'observations' and thus the future sea-level rise may become larger than those projected by climate models.*

Yes – we mostly agree with this summary. Importantly, we stress that we are aware that the discrepancy between the historical and projected sensitivities cannot be fully conclusive as it is comparing the response in two different periods. Hence the phrasing "*may become*" in the comment above.

It is correct that GMST refers to the global mean surface temperature. We define this in the data and methods section.

Minor disagreement: We would not call the AR5 and SROCC sea level projections, "*climate simulations from the CMIP5 model ensemble*". We note that the SLR projections in AR5 and SROCC is not projected directly by models, but rather using an afterburner to the models providing climate change projections.

**Revision plan**

- Check if it makes sense to move GMST definition into the main body of text.
- Ensure careful phrasing of the conclusions. We do not want to overstate the significance of "the discrepancy" between past and future. But we will emphasize the caveats related to the use of GCM climate projections further processed to get SLR information.

*Recommendation:*
*This is a surprisingly short manuscript, which in my view leaves many technical detailed unclear. It does not have a result section, and so it was for me difficult to interpret what the sole figure 1 and the sole table 1 is actually representing. The very concept of transient sea-level sensitivity requires a much deeper physical discussion. My impression is, therefore not positive. The manuscript seems in many respects to be incomplete.*

We plot published data in a deliberately provoking way, with minimal analysis. We strongly feel that the content is best suited for a short discussion letter rather than a long research article. Naturally, we are not satisfied that our condensed presentation apparently was unclear, and we will strive to improve that in an expanded revised version.

Figure 1 demonstrates that the transient sea-level sensitivity metric does captures most of the future model response. The IPCC assessments summarize our process knowledge. This is in our opinion a much stronger argument than physical discussions of how we might expect the system to respond to warming.

The primary objection seems to be that there may be physical mechanisms that could explain why the sensitivity of the 21st century would be different from during the historical period. I.e. there could potentially be an explanation for the discrepancy highlighted by figure 1. We want to stress that we absolutely do not assume that TSLS is constant through time. This is why we originally said: "*This does not automatically demonstrate a bias in model projections, but as a*
*minimum call for a detailed explanation*", and "*Future TSLS may well be different from the past,*
*…*". We will stress this even further in the revised manuscript.

**Revision plan**:

• Expand description of figure.
• Explain statistical methods in detail.
• Stress that sensitivity may be different in future from past, and that this can possibly
explain "the discrepancy" and assess the involved physical mechanisms more clearly.
• Elaborate substantially on the limitations of the metric.

*1) The definition of sea-level climate sensitivity, although used in some previous studies, is at*
*least rather questionable, and it was clearly questioned also in the AR5 report itself. This*
*manuscript should at the very least justify in the first place why this concept is meaningful.*

The AR5 questioned a universal linear relationship between sea level rise rate and temperature,
and therefore questioned projections based on extrapolations of the historical relationship. We
are fully aware of this and agree to this premise.

The main argument was that there may be physical reasons that cause future response to be
different from the past. Some mechanisms could cause the response to be non-linear, and other
mechanisms invoke non-stationarity where the sensitivity depend on the state of the system.
Figure 1 shows that the process-based models actually do show a near-linear response. A
linearization clearly captures most of the future response. This demonstrates that the TSLS
concept is meaningful as far as our process knowledge is concerned. Figure 1 therefore directly
eliminate a whole class potential problem raised in AR5.

Non-stationarity is another reason that future sensitivity may be different from the past. This
could cause the TSLS of the 20th century to be different from that of the 21st century. We fully
acknowledge this, and this is the main limitation of the comparison to historical data. On the
other hand, experts align much better with the extrapolations than the AR5/SROCC projections.
Nevertheless, we acknowledge that this is an issue and this is why we are generally careful to
not overstate the implications of the discrepancy. E.g. we write: "*This does not automatically*
*demonstrate a bias in model projections, but as a minimum call for a detailed explanation*".

**Revision plan:**

• Address explicitly the premises adopted in AR5 (and implicitly in SROCC) that a
universal linear relationship between sea level rise rate and temperature is
questionable.
• Stress that sensitivity may be different in future from past, and that this can possibly
explain "the discrepancy" and assess the involved physical mechanisms more clearly.
• Discuss non-linearity and non-stationary.
• Emphasize more strongly the limitations of the comparison to the observational
estimate. Especially in abstract.

*For instance global mean sea-level rise is brought about by two very different mechanisms:*
*expansion of the water column and melting of land ice. A backof-the-envelope calculation yields*
*that the global sea-level rise caused by the capture of an energy flux of 1 w/m2 by the liquid*
*ocean, and its subsequent expansion, is about 1.9 mm. This is very different from the sea-level*
*rise caused by the capture of of 1 w/m2 by land-ice and subsequent melting, assuming the ice is*
*already at 0C, (94mm). Of course, this also depends on where the heat flux is captured and many*
*regional details, but the difference between 1.9mm and 94mm is in principle enormous.*
*Therefore, the very concept of a linear relationship between energy flux imbalance and the rate*
*of global sea-level rise is physically questionable, at least it requires a plausible justification, as*
*the 'sensitivity' depends on the relative contribution of thermal expansion and melting.*

First, we agree that how the energy is spent will have a huge impact on the TSLS. But this just
illustrates that the TSLS metric quantifies an important aspect of the sea level response.

The idea of a linear response may be surprising or '*physically questionable*', but it is simply a fact
that the IPCC process-based projections have a 21st century response that is almost perfectly
linear in warming. Figure 1 demonstrates that.

The main objection hinges on a common misconception. The reasoning seems to go as follows:
Since we know that the relative proportions of ice melt and expansion are changing, and that
melt and expansion may have very different sensitivities, then the combined sensitivity (TSLS)
must be changing over time. However, this simply does not follow. Even in a model where every
contributor responds linearly to warming the relative proportions can change. This is
demonstrated in Note R1 in the end of this document.

Finally, the TSLS concept is just a metric that characterizes the first order response at a given
point in time. You can always linearize the response and talk about the slope. This is essentially
all we are doing. The concept does not require that the response is perfectly linear, nor does it
hinge on the relationship being stationary in time. Non-linearity and non-stationarity would of
course place limitations on how the metric can be used.

Minor note: There is a mismatch of units in your back-of-the-envelope calculation. The energy
capture is given as a rate, but time is missing from the corresponding sea level rise. Should it be
per mm/year?

**Revision plan:**

• Stress that we do not expect TSLS to be constant over time.
• Elaborate substantially on the limitations of the metric.
• Consider discussing common misconception.

*This contribution is rather uncertain for the future, but it seems to me clear that in the near*
*future melting will play a much bigger role through glacier melting, then perhaps a smaller role*
*as glaciers are completely melted and then again a bigger role when melting in Greenland and*
*Antarctica sets in. So it is really difficult for me to envisage a simple linear relationship to*
*describe this dependency. It may be that in practice it works, but this needs to be justified.*
*Unfortunately, I do not see which data could be used to justify this assumption. The centennial*
*smoothing assumed in this study would require several millennia of data for a robust*
*justification.*

The TSLS concept does not rely on a perfectly linear relationship. It is useful if a linearization is
a reasonable approximation of the relationship. We show that it in practice works for AR5 and

SROCC models. But it is still a simplification, -just as transient climate response is only an
approximation to how surface temperature respond to radiative forcing.

The objection here seems to be that there could be processes that change the sensitivity over
time. We do not assume that the sensitivity is constant, and especially not over several
millennia. We do, however, compare the historical sensitivity to the projection sensitivity. But
that is just a comparison. We note that there is a disconcerting discrepancy, and that this needs
to be explained. We may speculate that perhaps this is because the sensitivity has changed from
the 20th to the 21st century. But that would only be speculation without further study. One way
to address this would be to verify that the models used for projections can reproduce the sea
level rates of the historical past. Unfortunately, the aggregate sea level models used in SROCC
and AR5 have never been validated in this manner. We argue that the type of comparison we are
doing in this paper is the next best thing. The discrepancy to observations is disconcerting.

**Revision plan:**

• Stress that sensitivity may be different in future from past, and that this can possibly
explain "the discrepancy" and assess the involved physical mechanisms more clearly.
• Emphasize more strongly the limitations of the comparison to the observational
estimate. Especially in abstract.
• Call for historical validation of models used for sea level projections. Not just of the
individual contributor models, but also of the aggregate model.

Finally, it is a common misunderstanding that the total sensitivity must be changing because the
relative contributions of contributors are changing. However, it is perfectly mathematically
possible that the relative contributions change even if every contributor responds with a
constant linear sensitivity. This just requires that each component is not equally close to being
in balance. [See note R1 in the end of this response].

*2) Related to point 1, the CMIP5 global climate models do not include land ice melting. This is*
*the reason why the IPCC AR5 included a contribution to estimated sea-level rise by expert*
*knowledge. But I wonder how the comparison between AR5 models and observations can be*
*meaningful, when one of the key components is missing in the models. Therefore, it is not really*
*surprising that the sensitivity estimated from models is smaller than that estimated from*
*observations. This is again the reason why the IPCC augmented the estimated sea-level rise by*
*2100 with an approximate contribution from land-ice melting.*

We agree that the way ice contributions was treated in AR5 explains why AR5 has a too low
slope. We also agree that it is not surprising, and we already explain this in the manuscript, so it
is unclear what else we should do here.

Minor disagreement: AR5 did in fact include land-ice melting. It was only the dynamic
contribution where they used an approximate contribution based on expert knowledge.

*3) The approach in this manuscript seems rather similar to the approach by Rahmstorf (2007).*
*The reader would like to know in what aspects both approaches differ, and how this difference*
*my affect the results.*

The most important difference is that we are not making a projection, and we are not assuming
that that the future sensitivity will be like the past. This is an important distinction.

Rather,

• We define the TSLS metric.
• We demonstrate that the TSLS captures most of the 21st century response in AR5 and
SROCC. I.e. we address some of the concerns raised in the AR5 in response to semi-
empirical models such as Rahmstorf (2007).
• We compare the observational sensitivity to the projection sensitivity, and highlight a
disconcerting disagreement. We then "*call for a detailed explanation*".

Statistically there are also differences. Rahmstorf (2007) was criticized for assumptions
concerning statistical independence, and degrees of freedom. We avoid these issues by relying
on a single average sea level rate for each observational record. It is simply a better assumption
that the TG and SAT rates are independent. However, a drawback is that we are left with only a
few points to base our observational estimate of TSLS. Less data usually results in larger
uncertainties. A more detailed time series analysis of the tide gauge record could potentially
provide a TSLS estimate with tightened uncertainties (which would only make the numbers in
table 1 even more significant). However, the statistical assumptions of such an analysis would
be much more critical. Given the robust push-back here, then we are happy with our choice to
use a simple but rock-solid approach for our uncertainties.

Another motivation to using long-term values for the TG or SAT rate is that these are published
by the authors of the records. This means that figure 1 just is what it is. The location of the
points does not rely on any analysis we make.

**Revision plan:**

• Stress that sensitivity may be different in future from past, and that this can possibly
explain "the discrepancy" and assess the involved physical mechanisms more clearly.
• Emphasize more strongly the limitations of the comparison to the observational
estimate.
Explain that we only use published estimates, and motivation.

*4) I struggle to understand what Figure 1 and Table 1 are exactly showing ? Certainly the*
*caption or the main text should include a much lengthier description. Points that remain unclear*
*to me are: what is the averaging window (100 years as suggested in the main text?) If yes, the*
*global mean temperature observations would be just 1 point ?), What does the point labeled as*
*Sat9 represents ? Probably it represents the data in the satellite era, but there is no mention of*
*this in the main text, only one paper listed in the reference list. The same can be said about TG7.*
*To be honest, at this point I wonder whether the authors have carefully checked the manuscript*
*before submitting.*

We are of course not satisfied that our captions are not sufficiently clear, and we will work to
clarifying this in our revisions.

We regret overlooking the numbered references left in the figure from a prior version of the
manuscript. This will be fixed in the revision.

**Revision plan:**

• Expand caption – Explain what each point is, including their time span.
• Remove superscripts from figure.
• Explain more in main text time period.

*In the case of observations, if my interpretation is correct, the linear fit is constructed using two*
*points, both with different characteristics (one represents centennial means, the other satellite-*
*era means). Is linear fit with just two points enough to be extrapolated ? The extrapolation*
*would be even more questionable when considering that the physical processes would change*
*over time, as explained in my point 1. How were the uncertainties calculated considering that*
*the errors in each of these data points are different ?*

We acknowledge that the statistical details were not described in detail in the manuscript. We
will revise the manuscript with a more thorough description of the statistics. The extrapolation
is just a visual comparison, and should not be taken as a projection.

The observational fit is calculated using three
points:

[Figure]

• SAT: 1993-2017
• TG: 1900-1990
• PI: 1850-1900

The time intervals were chosen because this is
what was provided by the cited studies. All
points have their own uncertainties in both the
x and y directions (where x:T; y=SLRate). The y-
uncertainty is given in the cited studies, and the
x uncertainty was extracted from the HADCRUT4 ensemble for the same period.

In this paper we take uncertainties in both of the displayed variables into account. We do that
using Monte Carlo sampling. We make 10000 linear regressions, where each displayed variable
are perturbed according to their uncertainties. This gives an ensemble of slopes and intercepts
that we can extract statistics from. We report a TSLS based on these data and of 0.40
m/century/K [0.35-0.44].  For comparison standard weighted least squares regression (which
only takes errors on the dependent variable - typically chosen to be on the vertical axes - into
account) yields a substantially narrower uncertainties for the TSLS of 0.39 m/century/K [0.37-
0.41].

**Revision plan:**

• Explain statistical methods in detail.
• Stress that extrapolation is not a projection but plotted for comparison.
• Stress that sensitivity may be different in future from past, and that this can possibly
explain "the discrepancy" and assess the involved physical mechanisms more clearly.

*Further points*

*5) The main text mentions reconstructions of sea-level in the preindustrial period, but were have they been used? There is no mention of temperature reconstructions that could be used for the estimation of sea-level sensitivity.*

In the methods section we define the pre-industrial (PI) as 1850-1900 (following AR5). For that period we have an average temperature from HADCRUT4, and a sea level rate from Kopp et al. (2016). This is plotted as PI in figure 1. This point is used together with TG and SAT in the observational estimate of TSLS.

**Revision plan:**

State time-intervals in figure caption.

*6) The caption of the table mentions a level of significance in the difference of the sea-level sensitivity. How has it been calculated ?*

We realize that we did not detail that it we used a two-tailed test and the assumption of normality. We will add this in the revision.

To be 100% clear we also have an expanded explanation here:

We want to look at the difference between $TSLS_{AR5}$ and $TSLS_{obs}$. But these numbers are uncertain, and we want to know if that difference is large considering the uncertainties in both estimates. E.g. We want to look at the difference between $TSLS_{AR5}$ and $TSLS_{obs}$. But these numbers are uncertain, and we want to know if that difference is large considering the uncertainties in both estimates. For gaussian errors standard uncertainty of the difference will be the $\sigma^2_{\text{difference}} = \sigma^2_{\text{obs}} + \sigma^2_{\text{AR5}}$. Then the p-value can be looked up in the CDF of the normal distribution. This is basically a particularly simple t-test. In order to make the test we need the standard errors. There is a one to one relationship between standard error and likely range as we have assumed normality (the conversion factor is 1.048).
* * *
**Example calculation** (comparison between $TSLS_{obs}$ and $TSLS_{AR5}$).
From table 1 we have:
$TSLS_{obs} = 0.391$ and $TSLS_{AR5} = 0.274$
$\sigma_{obs} = (0.391 - 0.349) \cdot 1.048 = 0.044$
$\sigma_{AR5} = (0.303 - 0.274) \cdot 1.048 = 0.030$

This yields:
$\Delta TSLS = 0.391 - 0.274 = 0.117$

$\sigma_{difference} = \sqrt{\sigma^2_{\text{obs}} + \sigma^2_{\text{AR5}}} = 0.053$

The probability of values greater than 0.117 in a normal distribution with zero mean and that $\sigma_{difference}$ is p=0.013. That is the p-value of a one-tailed test. The two-tailed probability will be twice as high. This is the p-value we report to be below 0.05 in table 1.
* * *
**Revision plan:**

- Write that it is a two-tailed test assuming normality.

| *7) The temperature anomaly are referred to the base line 1986-2005. What is the reason for this*
| *short base line, when the link between T and sea-level rate is assumed to be at centennial scales*
| *? It does not seem consistent. I guess there is an explanation for it, but the manuscript is so short*
| *and concise that the reader is left wondering.*

Here, we simply adopt the baseline from the IPCC reports. This choice of base line is just a
translation of the plot and has no impact on the slope (TSLS) or the 'discrepancy'. By adopting
the same baseline as IPCC, we avoid introducing additional uncertainty by redefining the
baseline. This means we can plot the AR5 and SROCC values exactly as reported. We actually
write: "*We follow AR5 (Church et al., 2013) and use a 1986-2005 baseline for temperature*
*anomalies ...*".

**Revision plan:**

• Be explicit about baseline motivation.

| *The latter are just examples of open technical questions that should be clear in a properly*
| *formatted manuscript, with proper length*

We hope to address all the technical questions following the plan outlined in the answers above.
The revised manuscript will also be more explicit about the limitations of the TSLS metric and
the comparison between past and future. This will result in a longer text, but we still aim for a
letter format.

**Note R1: Changing proportions, yet constant sensitivity**

The sea level budget is changing, and we expect ice sheet melt to increasingly dominate the budget. This might lead one to argue that the sensitivity must be changing as we don't expect the individual contributors to be equally sensitive to warming. In this section we present a case for why that is a flawed argument. We show that even in a completely linear model the relative proportions of the individual sea level contributors can change.

Let's assume for the moment, that the rate of sea level rise is just the sum of the contribution from ice melt ($\dot{M}$) and the contribution from thermal expansion ($\dot{E}$). We write:

$$\dot{S} = \dot{M} + \dot{E}$$

Let's also assume that these two contributions respond linearly to warming.

$$\dot{M} = a_M T + b_M$$

$$\dot{E} = a_E T + b_E$$

We insert and get a linear model for the sea level rate:

$$\dot{S} = (a_M + a_E)T + b_M + b_E$$

The proportion of sea level rise due to ice melt becomes

$$\frac{\dot{M}}{\dot{S}} = \frac{a_M T + b_M}{(a_M + a_E)T + b_M + b_E}.$$

This is not generally constant in T. This demonstrates that a changing proportion of ice melt does not necessarily imply a changing sensitivity to warming.

**Response to RC3: Anonymous referee #3**

*In this manuscript, the authors define the new concept of transient sea level sensitivity that is inspired by the transient climate sensitivity but that is adapted to the sea level problem. In particular it relates the sea level rise over a century with the average temperature anomaly compared to a steady state over the same period. I think this concept, even with all its drawbacks, has the potential to be useful but the arguments developed in this manuscript needs to be further developed to be convincing. Especially since the authors make important claims about the underestimation of future sea level rise by the IPCC AR5 and SROCC process-based method.*

We agree that there are limitations, and are convinced that the TSLS will be a useful tool.

**Revision plan**

- Elaborate substantially on the limitations of the metric.

*General comments:*

*An important motivation to define the TSLS is the linear relationship between sea level change and GMST in both observations and models. However that relationship is not very convincing. I agree with the theoretical points mentioned by referee #2 so I will not come back on those but I will focus on the observations and model data used in Figure 1:*

The monotonous relationship with almost no scatter is an important justification. To be useful it does not have to be a linear relationship, but a linearization has to be a reasonable approximation that characterizes most of the response. It will be an approximation, and you should be careful extrapolating. Different time periods can have different sensitivity, and we expect the response to become non-linear for intense warming scenarios (as seen in SROCC). Admittedly, this we can be more explicit about in the text.

We hope that you will take a look at our responses to referee #2, and check if we have addressed the concerns you share.

*1) The observational data used here to back up such a relationship is weak. There are only three points, moreover the pre-industrial and tide gauge periods are very close to each other. With therefore the main point driving the slope of the linear relation being the satellite period which is only around 25 years. I would suggest that if the author think 25 years is enough to estimate the TSLS then the tide gauge period could be split in a few 25 years periods.*

We agree that it is probably possible to make a better estimate of the historical TSLS, using a more sophisticated statistical analysis of the full historical data. However, this is not trivial. E.g. it is important to take uncertainty autocovariance of the tide gauge record properly into account. There are multiple reasons why we decided to restrict our analysis to published estimates rather than our own statistical analysis of the tide gauge record:

- We are writing a short letter that may be seen as controversial by some. It seems more appropriate and more convincing to use published estimates, rather than making a highly technical statistical analysis with lots of assumptions, which would seem to add to the controversy.

•   We are convinced that this approach yields conservative uncertainty estimates.
•   It is a better assumption that the three historical estimates are independent, than if you
  slice the tidegauge record into shorter sections, in which case they definitely will not be.
•   Downsampling of the tide gauge record has been done before (e.g. Rahmstorf 2007). It
  sparked criticism of the statistical assumptions, which is key to us to avoid.

Finally, our TSLS estimate should not be the final word on the subject – we want to add a new
element into the assessment of all available information about sea level rise information.

The historical TSLS is estimated using data from 1850-2017. It is correct that the shortest slice
of data is the altimetry record which is only ~25yrs. It is, however, also the least noisy.

We disagree that pre-industrial and tide gauge rates are close. The pre-industrial rate is
centered around 1875, and the tide gauge rate is centered at 1945.

**Revision plan**

Explain that we only use published estimates, and motivation.

*2) For model data the uncertainty lines are obtained from the assumption of full covariance*
*between GMST and sea level uncertainties in IPCC projections. But that is not the case at all,*
*there are many sources of uncertainty in the sea level projection that are independent of*
*temperature. For example Greenland and Antarctic ice dynamic contribution, glacier model*
*uncertainty (four different models are used in AR5 and SROCC). The assumption is justified by*
*the fact that when it is made it shows a linear relationship between GMST and sea level but this*
*is what the authors try to demonstrate. Also for SROCC the linearity doesn't seem to hold at all.*

First, we want to emphasize that the near-linear relationship in the models is demonstrated by
the central estimates alone. So, we do not see the point of claiming a circular argument here.

It is unfortunately so that the IPCC reports offers very little information that can be used to infer
the uncertainty covariance. We know that the process based models for the ice contributions
are directly driven by temperatures in AR5 (see sections 13.SM.1.3 – 13.SM.1.5). So, a priori we
know that any uncertainty in temperature will be directly reflected in the modelled rate. I.e. we
know there will be a high degree of uncertainty covariance. We chose to go with the simplest
assumption: full covariance. We did, however, look into an alternative method of estimating
covariance.

The IPCC reports provides us with central estimates and a likely range. We can frame that as
$T \pm \sigma_T$ and $\dot{S} \pm \sigma_{\dot{S}}$. So, we know the uncertainty ellipse has to fit inside a rectangle with
width=$2\sigma_T$ and height=$2\sigma_{\dot{S}}$. Knowing how the ellipse is oriented inside the box is equivalent to
knowing the uncertainty covariance matrix. From the central estimates we have some idea of
how sea level rate depends on temperature. In lack of better information, it seems reasonable to
assume that one axis of the uncertainty ellipse should be aligned with the curve between central
estimates. From figure R3.1 we see that when the line between central estimates approaches the
corner of the rectangle (panelA) then we have a situation that approaches full covariance. This
is almost exactly the situation we have in figure 1 in the manuscript. The high and low end
estimates fall on the same curve as the central estimates. Notice: You can see that if you have a
situation like figure R3.1B then the top right corner of the red box would fall above the line
between central estimates. If we use this more complicated approach outlined here, then we
estimate uncertainty correlation coefficients of more than 0.95 for both AR5 and SROCC

(derived from the uncertainty covariance matrix). We decided to not use this approach to derive
the covariance matrix because:

• Need to assume symmetric gaussian errors.
• Need to assume a "local" linear relationship. (Not great for SROCC).
• Impossible to avoid assumption concerning how to orient the ellipse.
• It is rather complicated to explain.

In short, we prefer to keep the imperfect "full covariance" assumption. It is much simpler, and
can better deal with non-linearity. These principles, we wish to make more explicit.

[Figure]

Figure R3.1: A central estimate (black dot) with associated uncertainties $\sigma_T$ and $\sigma_{\dot{S}}$ (red). The uncertainty covariance matrix is represented by an ellipse. The ellipse must be inscribed in the rectangle. If we assume that the one axis of the ellipse is aligned with the curve between central estimates, then we can infer the ellipse parameters (=the covariance matrix). The panels compare two situations: One where the curve between central estimates curve nearly hit the corner of the uncertainty rectangle (panel A), and one where it does not (panel B). In panel A the ellipse approaches full covariance.

**Revision plan**

• Add a more complete description that explains that full covariance is unlikely, and how
it impacts results.

*l.47: "This does not automatically demonstrate a bias in model projections, but as a minimum*
*call for a detailed explanation."*
*Since this is the main claim of this short paper I think attempting to provide an explanation falls*
*on the shoulders of the authors. There is already some literature on that subject see for example*
*Slangen et al. 2017, in particular section 4:*

*"When all the contributions are combined, the models add up to a GMSL change of 92 6 47mm*
*for the period from 1901–20 to 1996–2015 (Table 4, Fig. 9a). Compared to the average of the*
*four reconstructed global mean time series for the overlapping period from 1901–20 to 1988–*
*2007 (Table 5, Fig. 9a, the model simulations clearly underestimate the observed GMSL and*
*explain only 50% 6 30% of the observed change (using 61.65s of the models to the mean of the*
*observations)."*

*And the following discussion on adding corrections to the sea level computed from the models to*
*solve the issue.*

Thank you – this is useful context.

It is important to note the context that AR5 and SROCC does not provide their own hindcasts
using the same process-based model used for projecting sea level rise. I.e. the aggregate
projections are not adequately validated against the historical record. It is very disconcerting
that there is a discrepancy between the historical response and models of the future. Slangen et
al. (2017) is really useful here.

The main contribution is in our opinion the concept.

**Revision plan:**

- Discuss Slangen2017 as context.
- Stress even more limitations of a comparison between two different periods: historical
and projections.

*Small comments:*

*- Figure 1: I can't find an explanation for the numbers in PI11, TG7, Sat9 and others.*

This was a leftover from an early version of the manuscript. This will be removed, and the
caption expanded.

*Slangen, Aimée B. A., Benoit Meyssignac, Cecile Agosta, Nicolas Champollion, John A. Church,*
*Xavier Fettweis, Stefan R. M. Ligtenberg, et al. "Evaluating Model Simulations of Twentieth-*
*Century Sea Level Rise. Part I: Global Mean Sea Level Change." Journal of Climate 30, no. 21*
*(November 2017): 8539–63. https://doi.org/10.1175/JCLI-D-17-0110.1.*

**Response to RC4: Anonymous Referee #4**

*The paper The transient sensitivity of sea level rise by Grinsted and Christensen discusses the relationship between global mean surface temperature and global mean sea level rise on a time scale of the order of a century. The authors acknowledge earlier work on the topic and frame the relation between temperature and sea level rise as an independent proxy for the evaluation of recent assessments of sea level rise projections that are biased low compared to observations. The article claims a linear sea level sensitivity of 0.4 m/century/K based on observations and either lower sensitivity in AR5 or higher balance temperature in SROCC and Bamber et al., 2019., respectively.*

*General comments*

*The paper is very short and concentrates on the discussion of the discrepancy between the parameters of linear regressions between averaged global mean surface temperature and global mean sea level rise, based on observations (past) and climate projections (future). In the face of high and rising stakes on the response to sea level rise additional proxies for the evaluation of projections of sea level rise are needed. The paper contributes to this end in bringing back the sea level sensitivity into the discussion. I think it is worth to be published and discussed in the community. The paper misses the opportunity to go deeper into the matter and offer thoughts or strategies how to address the discrepancies in transient sea level sensitivity between observations and climate projections.*

Thank you. We agree there are limitations to the metric, but also that it is serves as a useful reality check on sea level models – the comments here indicate that we may expand on the underlying ideas. In particular, we gather from the full set of reviews that we need to discuss limitations more.

**Revision plan:**

- Add more to the motivation part of the manuscript
- Elaborate substantially on the limitations of the metric.
- Add an outlook for how TSLS discrepancies can be addressed.
    - Brainstorm to consider when revising:
    - Ensure that projection models also have hindcasts of the historical past.
    - Look into the transient sensitivity of individual contributors.
    - Understand how TSLS changes over time
    - Model studies to understand the limitations of TSLS.

*Specific comments*

*I wonder whether we could learn something more about the impact of model development if the current analysis would include older projections like AR3 and AR4. Those were already below GMSL rise according to Rahmstorf 2007, Horton et al. 2008.*

It is a great idea to look into the TSLS of sea level models used in past IPCC reports (including FAR and SAR to complete the picture). However, this is a distraction and beyond the scope in this manuscript.

*One weak point of the analysis, as I see it, is the different ranges of GMST used for the regressions of the observations and model projections. Would it be possible and useful to include model estimates from paleo runs that had GMST anomalies in the same range as those projected for the 21st century?*

We agree that this is an important limitation, and based on the full set of reviews we also realize that we need to more explicitly discuss this limitation. Unfortunately, there is very little we can do about it as AR5 and SROCC has not published hindcasts with the same models used for hindcasts.

**Revision plan:**

- Stress that sensitivity may be different in future from past, and that this can possibly explain "the discrepancy" and assess the involved physical mechanisms more clearly.
- Expand the discussion and emphasize more strongly the limitations of the comparison to the observational estimate.

*The regression lines in Fig. 1 should pass through the mean time-averaged GMST anomaly and the mean sea level rate. Is there any information contained in the scatter of the mean GMST and mean GMSL rate of the individual regressions?*

For the observational trend, and the AR5 trend then the scatter around the trend line is so small compared to the uncertainty of the individual points that I would be careful to read anything into this. However, SROCC responds more non-linearly and there is deviation a straight line fit. The sensitivity is clearly increasing with warming. The TSLS we report is an average over the range of scenarios plotted.

If you were asking for more details about the statistical procedures, then please take a look at our replies to reviewer 2.

*It would be interesting to discuss some of the physical processes, thresholds, time scales and limitations, that would render the relationship between averaged GMST and GMSL rate non-linear. It would help to establish the transient sea level sensitivity as a metric next to equilibrium sea level rise on longer time scales.*

We agree.

**Revision plan:**

- Stress that sensitivity may be different in future from past, and that this can possibly explain "the discrepancy" and assess the involved physical mechanisms more clearly.
- Emphasize more strongly the limitations of the comparison to the observational estimate. Discuss time scales.

*Are current climate models or model ensembles good enough so that their uncertainty in GMST was smaller than the uncertainty in balance temperature in Table 1? Is the spread in balance temperature inherent in climate models or does it come from the combination of climate models (GMST, steric) with process models (ice sheets dynamics)?*

The answer here will be a little speculative, and so we have not added it to the manuscript. I
believe the uncertainty in balance temperature is a consequence of the long equilibration time
scales for several of the contributors. It requires a long spin-up of both the ocean and the ice
sheets to ensure that it has the full memory of the long term forcing. This will be reflected in the
model balance temperature. It will also put strong demands on the long term forcing. We will
consider to add a short paragraph on this in the discussion, if it helps to reassure other parts of
our discussion.

**Revision plan:**

• Consider adding a short speculative paragraph on uncertainties in balance temperature.

*From Table 1 one could deduce sea level rise of 0.28, 0.05, 0.17 and 0.17 m/century at balance*
*temperature. The 0.28 m/century sea level rise in the observations at balance temperature is*
*already above the 0.1-0.2 m/century sea level rise for the 20th century. Since sea level rise is*
*accelerating we are probably above balance temperature since at least the satellite era. This*
*seems to point to a contradiction in the data and the assumption of a linear process. How can*
*the balance temperature be interpreted or how well can we know it?*

There appears to be some confusion with the meaning of the terms "balance temperature" and
the "baseline temperature". The quoted numbers (0.28 etc.) are the sea level rate at T=0,
calculated as $\dot{S} = TSLS \cdot (0 - T_{balance})$. So, the 0.28m/century is the sea level rate when
temperature is equal to the baseline temperature reference (rather than "at balance
temperature").

This may seem like a minor point but: we would disagree that we assume a linear process.
Rather we argue that a linearization is a reasonable approximation to the response. There are
limits to how far that linearization would work, but that does not mean that TSLS is not useful. It
just means that the state of the system can change so much that the sensitivity to warming
changes.

*Technical corrections*

*l6: assessments from the Intergovernmental Panel on Climate Change implies*
*l20: and melts.. A perturbation*
*l20: perturbation in greenhouse gas concentrations change*
*l47: table 1 and figure 1*
*l52: table 1*
*l63: figure 1*
*l69: figure 1*

We have checked the lines mentioned, but we cannot understand what technical corrections the
referee has in mind. These specific lines look good to us.

**The transient sensitivity of sea level rise**

Aslak Grinsted[1] and Jens Hesselbjerg Christensen[1,2]

[1]Physics of Ice, Climate and Earth, Niels Bohr Institute, University of Copenhagen, Denmark.
[2]NORCE Climate, Bergen, Norway

*Correspondence to*: Aslak Grinsted (aslak@nbi.ku.dk)

**Abstract.** Recent assessments from the Intergovernmental Panel on Climate Change (IPCC) imply  that global mean sea level is unlikely[1] to rise more than about 1.1m within this century, but will increase  further  beyond 2100, even within the most intensive future anthropogenic greenhouse gas emission scenarios are higher levels assessed to be unlikely. However, some studies conclude that considerably greater sea level rise could be realized, and a number of experts assign a substantially higher likelihood of such a future. To understand this discrepancy, it would be useful to have scenario independent metrics that can be compared between different approaches. The concept of a transient climate  sensitivity has proven to be useful to compare the response of climate models. Here, we introduce a similar metric for sea level science. By analyzing mean rate of change in sea level (not sea level itself), we identify a near linear relationship with global mean surface temperature (and therefore accumulated carbon dioxide emissions) in both model projections, and in observations on a century time scale. This motivates us to define the 'Transient Sea Level Sensitivity' as the increase in the sea level rate associated with a given warming in units of m/century/K. We find that model projections fall below extrapolation based on recent observational records. This comparison  suggests that the likely upper level of sea level projections in recent IPCC reports would be too low.

**1 Introduction**

Our planet is warming as anthropogenic emissions are increasing the atmospheric concentration of carbon dioxide. This warming causes sea levels to rise as oceans expand and ice on land melts. A perturbation in greenhouse gas concentrations changes the balance of energy fluxes between the atmosphere and the ocean surface, and  the balance of mass fluxes to and from glaciers and ice sheets. However, the oceans and ice sheets are vast and it takes centuries to heat the oceans, and millenia for ice sheets to respond and retreat to a new equilibrium (Clark et al. 2018; Li et al., 2013; De Conto and Pollard,

2016; Oppenheimer et al. 2019; Clark et al., 2018). In this sense the ice sheets and oceans have a large inertia: An increase in forcing result in a long-term commitment to sea level rise. Simulations by Clark et al. (2018) indicate an equilibrium sea level sensitivity of ~2m/100 GtC emitted $CO_2$. The equilibrium sensitivity can be compared to paleo-data (e.g. Foster and
* * *
[1] The following terms are adopted by the IPCC to indicate the assessed likelihood of an outcome or a result: Virtually certain 99–100% probability, Very likely 90–100%, Likely 66–100%, About as likely as not 33–66%, Unlikely 0–33%, Very unlikely 0–10%, Exceptionally unlikely 0–1%.

[revised manuscript text omitted]

---

## Author Response (AR2)

| 1                          | Response to Reviewer #2                                                                                                                                                                                                                                                                                                                                                                                                                                                     |  |  |
|----------------------------|-----------------------------------------------------------------------------------------------------------------------------------------------------------------------------------------------------------------------------------------------------------------------------------------------------------------------------------------------------------------------------------------------------------------------------------------------------------------------------|--|--|
| 2
3
4
5           | I thank the authors for considering my comments on the initial submission. I think that
manuscript has been improved, but I still have some concerns, as I explained below. Some of
them might be a matter of style, and thus are debatable, but the authors and the editor may
want to think about them once more.                                                                                                                                                |  |  |
| 6                          |                                                                                                                                                                                                                                                                                                                                                                                                                                                                             |  |  |
| 7
8
9
10          | Improvements: The main message of the manuscript is now clearer. The study defines a transient sensitivity and attempts to estimate from observations and from model runs. These estimation shows a discrepancy, and the authors conclude that the models may be underestimating future sea-level rise.                                                                                                                                                                     |  |  |
| 11
12                   | ■ We agree that the manuscript has improved a lot based on your feedback. Thank you. We hope we have improved it further this time around and made the conclusion(s) even more clear.                                                                                                                                                                                                                                                                                       |  |  |
| 13                         |                                                                                                                                                                                                                                                                                                                                                                                                                                                                             |  |  |
| 14
15
16
17       | 1) One main concern is again related to the length of the manuscript, whereas at the same time it compresses important information too strongly. I do not really understand why the authors want to cram all that information, as the length of the manuscript is by far not close to usual limits.                                                                                                                                                                         |  |  |
| 18
19
20
21       | We agree that we are not pressed for space. We will see what we can add to accommodate your concerns. Therefore, we have expanded the manuscript considerably and now include a section that articulates some of the potential caveats related to non-linearity and non-stationarity that were raised in the discussion of the manuscript                                                                                                                                   |  |  |
| 22                         |                                                                                                                                                                                                                                                                                                                                                                                                                                                                             |  |  |
| 23
24
25
26
27 | For instance, the the main point of the study is the disagreement between the estimations of the transient sensitivity from observations and models. The reader would think that the methodology, including a clear description of the uncertainties, biases, etc. is crucial. Yet, the method section devotes just one sentence to describe the estimation of uncertainties (We use Monte Carlo sampling)- Well, Mane Carlo sampling can be accomplished in many different |  |  |

- 28 ways, for instance to preserve autocorrelation of the regression residuals, with replacement or 29 without replacement. The reader does not even know how many samples enter the regression - it
- 30 seems that just 3 data points (?). If this is true, how is Monte Carlo re sampling really 31 accomplished ?
- 32 It appears to us that this is the primary concern: That the referee would like a more expanded 33 description of the methods section in particular.
- 34 It really is very simple what we are doing in this Monte Carlo sampling, hence the brief
- 35 explanation. The number of observational/historical points is 3, as should be clear from the data
- 36 section, and figure 2. But we have no space concerns and so you are correct, it would be good to
- 37 state it in the methods section too. The three estimates come from different studies, and are well
- separated in time and it is therefore reasonable to assume the uncertainties are independent 38
- 39 which makes the MC sampling trivial. We have added more detail to make this clear. We have revised the description in the methods text accordingly:
- 40
- 41 "The relationship between temperature and GMSL rate is estimated for each group of points using 42 linear regression. The three observational estimates of both temperature and sea level rate

- 43 (Figure 2, black) are uncertain. We take the uncertainties to be independent as the three estimates
- 44 are sourced from separate studies using different data sources, different methods, and are well
- 45 separated in time. We assume independent gaussian errors which we propagate to our estimates
- 46 of the line parameters listed in Table 1 using Monte Carlo sampling."

- 48 Just to be 100% clear in this reply then here's a verbose description and a code block (Python)
- 49 that you can compare to the above description. We have three x-y pairs, each with their own  $\sigma_x$
- and  $\sigma_y$  errors. We treat these errors as independent and gaussian. The source for the  $\sigma_y$ -values
- are the same three publications that provide the y-values (I.e. the sea level rate estimates). The
- $\sigma_x$  errors are obtained from the HadCRUT4 ensemble as described in the data section.

53

- 54 Note that we have compared the above MC derived uncertainties to traditional weighted least
- 55 squares regression (using statsmodels.regression.linear\_model.WLS). The results are virtually
- 56 identical. Also note: WLS also assume gaussian independent errors, but only allows for errors in
- 57 the y-direction.

- 2) Perhaps my most important concerns is indeed related to that very limited number of
  samples. The study claims that although the physical processes relating global mean
  temperature and sea level rise would in principle render that link non-linear,, the data indicate
  that the link is close to linear., at least in the range of the observed changes. I struggle to
  understand this claim when the available number of samples is just 3. ...
- 64 First: We do not just use the three historical points as an argument for linearity. So, the criticism 65 here seems somewhat misplaced. We observe that the models (AR5 and SROCC) show a nearlinear behavior. That is not just three points. In the case of AR5 we have a near perfect linear 66 67 relationship for 15 pairs of values (including the upper and lower ranges), and similar for the 9 68 SROCC points. We then also find that the observations are close to linear. That may not be very 69 impressive given that N=3, but this observation does not stand alone. Thus, we have three 70 separate groups of points, and all of them are near linear. For all groups of points the linear 71 correlation coefficients are greater than 0.98 (see table R1). Because of this we think we are 72 justified in saying: "We find that both model projections and observations show a near linear 73 relationship between century averaged temperature change and the average rate of sea level rise
- 74 (Figure 2)".
- 75 The relationship between the SL rate and temperature may not be completely linear. We agree,
- and we explicitly state this in the manuscript already. However, we presume that we can agree
- that the rate is highly correlated with temperature. The relationship might not be linear, but if it
- is reasonably close to monotonic, then it makes sense to make a first order linear approximation
- over a given range. And this is precisely what we are doing. We do not argue that the
- 80 relationship is linear. Indeed, we very explicitly note that it could very well be non-linear and

- 81 that there are limits to an extrapolation. But within those limits the linearization we perform
- 82 makes sense.
- 83 **Table R1:** Linear Pearson correlation coefficients of all the points in the historical, AR5 and
- 84 SROCC. For the AR5/SROCC values two separate correlations are calculated either including or
- 85 excluding the upper and lower likely limits. Correlation and p-values were calculated using
- 86 scipy.stats.pearsonr.

|                                | Ν  | Pearson Corr | p-value |
|--------------------------------|----|--------------|---------|
| Historical observations        | 3  | 0.9993       | 0.023   |
| AR5 (incl. upper/lower)        | 15 | 0.997        | 4*10-16 |
| AR5 (only central estimates)   | 5  | 0.994        | 0.0006  |
| SROCC (incl. upper/lower)      | 9  | 0.982        | 2*10-6  |
| SROCC (only central estimates) | 3  | 0.998        | 0.034   |

87

90 ... Even when looking at Fig 1, I would even go so far to say that a regression line drawn using

- 91 the first two observational data points, PI and TG would not hit the third data point Sat even 92 considering the uncertainty ranges
- 92 considering the uncertainty ranges.
- We perform this test in Figure R2. It turns out that the Sat estimate \*is\* consistent with the
   extrapolation of a line based on PI and TG alone.

| 96  | I wonder if the claim of linearity would hold in the perfect world of climate models. For instance,  |
|-----|------------------------------------------------------------------------------------------------------|
| 97  | if we also include the corresponding pre-industrial and 20th century data points derived from        |
| 98  | one one model run, would we see a linear relationship through the whole period from pre-             |
| 99  | industrial to 2100 ? If the linear assumption of this study holds, this should be the case. Actually |
| 100 | it should be the case for all individual model runs, as each run would be a surrogate for            |
| 101 | observations. If this assumption does not hold for the individual model runs - for instance if the   |

88 We have added some correlations to the results section and explicitly state N.

- the scenario data points fall bellow or above the regression line drawn with the PI and 20th
  century data points, then the linear hypothesis would not be correct, and the comparison shown
  in Figure 1 would not be indicative of an under or overestimation by the models.
- We also wonder what models would say for the past. At the moment it is impossible to do
   anything but speculate, because unfortunately there are no hindcasts with the same models
- used for projections and we very much encourage such studies. Here's how we frame this in the
   manuscript:
- 109 *"Ideally, we would test the models using hindcasts to verify their ability to reproduce the past."*
- 110 Unfortunately, such hindcasts are unavailable for sea level projection models assessed in both AR5
- and SROCC. This is critical as Slangen et al. (2017) identified substantial biases in hindcasts of
- 112 Greenland surface mass balance, glacier mass loss, and deep ocean heating. Adjusting for these
- systematic biases increase the modelled sea level rise over the 20th century by ~50%."
- 114 Note that we restrict our claim of linearity to exactly what we observe. We observe a linearity
- for the 21stC model response. We do not claim nor do we want to that this is universal and
- 116 necessarily leave open whether the same linearity could hold for the past. We explicitly discuss
- 117 how the relationship can be broken by non-linearities and non-stationarities.
- 118 We have added a new section called "reflections on the method". In this new section we discuss
- how comparisons between past and future TSLS, vs hindcasts and propose using past TSLS asan emergent constraint. We write:
- 121 "Sea level projections in the IPCC Fifth Assessment Report (AR5; Church et al., 2013), and the
- 122 Special Report on the Ocean and Cryosphere in a Changing Climate (SROCC; Oppenheimer et al.,
- 123 2019) are unfortunately not accompanied by hindcasts using the same model framework used for
- 124 projections. It is therefore impossible to verify that these models can reproduce historical sea level
- 125 rise. We can, however, compare the TSLS of model projections to the TSLS implied by historical
- 126 records, and this can serve as a reality check. We have to keep in mind that TSLS potentially can
- 127 change over time, and that a comparison between different periods cannot be as conclusive. We
- 128 therefore appeal that future sea level based on modelling are not only used for projections but also
- 129 include results based on model hindcasts. Ice sheets and ocean heat content has multi century
- 130 response times and this can lead to model drift if the model is not perfectly initialized. To inform
- about the future, it is therefore a necessity but not sufficient that model can reproduce the total sea
- 132 level rise over the 20th century. It is critical that sea level models also have sensitivities that are
- 133 compatible with observations. We therefore propose that the historical TSLS should be used as an
- 134 emergent constraint of sea level models."
- 135
- 136
- 137 This would be one possibility., perhaps there are others. What I mean here is that the manuscript
  138 leaves open some avenues to support or reject the main linear hypothesis, and I do not see why
  139 they are not pursued further in this study.
- 140 Looking into hindcasts is impossible because these model hindcasts do not exist. So
- 141 unfortunately, we will have to leave some questions open. We now draw attention to these open
- 142 questions and the importance of hindcasts in the both the conclusion and the new "reflections
- 143 on the method section".

144 Further we would argue that it is incorrect to consider the linear approximation a hypothesis,

- and an approximation is not a hypothesis that can be rejected. The linear approximation is
- 146 supported by the very high Pearson correlation coefficients (See table R1).
- 147
- 148
- 149 3) The linear hypothesis raises some additional points. ...

... The authors agree that the two main process causing sea level rise (thermal expansion and 150 151 land ice melting) have a vastly different temperature sensitivities. Then the question arises as to 152 why the linear link between temperature and sea level rise would hold. Is it because the sea level 153 rise is still too small to show that non-linearly? is it because of these two processes only one has 154 been dominant so far? There are estimations of the contribution of these processes to 20th 155 century sea level rise. Frederike et al 2020, (doi: /10.1038/s41586-020-2591-3) found that the 156 main mechanisms during the 20th century was land-ice melting and that that contribution has 157 grown larger through time. It is plausible to assume that for the preindustrial period the main mechanism was water expansion. Thus, why is the link still linear? 158

- 159 I am of course aware that these questions are not easily to solve, but why not include here a a160 first step ?
- 161

162 First, we want to stress that we do not have a linear hypothesis as already alluded to above. We

163 have a linear first order approximation. This is an important distinction, and in our opinion, this

also reduces the burden of proof drastically. We believe that our manuscript already makes this

165 very clear. E.g. "*Nature is complex and will be both non-linear and non-stationary, and this places*

166 *limits on extrapolation. Regardless, the sea level response can always be characterized using the*

167 TSLS metric, and we can compare and contrast different estimates. "

168 We acknowledge the shifting balance of the budget as nicely demonstrated by Frederikse et al.

169 2020. We also acknowledge that the different contributors have different sensitivities. However,

170 it is simply wrong to conclude that this must give rise to a non-linearity. Consider the

- 171 illustrative example in figure R2. Here, we have two contributors, both modelled as being
- 172 completely linear in temperature, yet in relative terms the sea level budget starts out being

173 dominated by expansion, but eventually ice melt takes over.